# Effective Graph Representation Learning via Smoothed Contrastive Learning

## Abstract

Graph contrastive learning (GCL) aligns node representations through the utilization of positive/negative node pairs, a selection process that typically relies on the correspondences and non-correspondences among nodes within two augmented graphs. The conventional GCL approaches incorporate negative samples uniformly in the contrastive loss, resulting in the equal treatment of misclassified false negative nodes, regardless of their proximity to the true positive. In this paper, we present a Smoothed Graph Contrastive Learning model (SGCL), which leverages the geometric structure of augmented graphs to exploit proximity information associated with positive/negative pairs in contrastive loss. The proposed SGCL adjusts the significance of these pairs in contrastive loss by incorporating three distinct smoothing techniques that yield smoothed positive/negative pairs. To enhance scalability for large-scale graphs, the proposed framework incorporates a graph batch-generating strategy that partitions the given graphs into multiple subgraphs, facilitating efficient training in separate batches. Through extensive experimentation in an unsupervised setting on various benchmarks, particularly those of large scale, we demonstrate the superiority of our proposed framework.

## 1 Introduction

Graph Neural Networks (GNNs) Gilmer et al. (2017); Kipf & Welling (2017); Xu et al. (2019b) have developed rapidly by providing the powerful frameworks for the analysis of graph-structured data. A significant portion of GNNs primarily focus on (semi-)supervised learning, which requires access to abundant labeled data Veličković et al. (2018); Kipf & Welling (2017); Behmanesh et al. (2023). However, labeling graphs is challenging because they often represent specialized concepts within domains like biology.

Graph Contrastive Learning (GCL), as a new paradigm of Self-Supervised Learning (SSL) Liu et al. (2023) in the graph domain, has emerged to address the challenge of learning meaningful representations from graph-structured data Wu et al. (2023); Xie et al. (2023). They leverage the principles of self-supervised learning and contrastive loss Li et al. (2019) to form a simplified representation of graph-structured data without relying on supervised data.

In a typical GCL approach, several graph views are generated through stochastic augmentations of the input graph. Subsequently, representations are learned by comparing congruent representations of each node, as an anchor instance, with its positive/negative samples from other views Veličković et al. (2019); Zhu et al. (2020); Hassani & Khasahmadi (2020). More specifically, the GCL approach initially captures the inherent semantics of the graph to identify the positive and negative nodes. Then, the contrastive loss efficiently pulls the representation of the positive nodes or subgraphs closer together in the embedding space while simultaneously pushing negative ones apart.

Conventional GCL methods follow a straightforward principle when distinguishing between positive and negative pairs: pairs of corresponding points in augmented views are considered positive pairs (similar), while all other pairs are regarded as negative pairs (dissimilar) Zhu et al. (2020). This strategy ensures that for each anchor node in one augmented view, there exists one positive pair, while all remaining nodes in the second augmented view are paired as negatives.

In contrast to the positive pairs, which are reliably associated with nodes having a similar semantic, there is a significant number of negative pairs that have the potential for false negatives. With this

strategy, GCL approaches allocate negative pairs between views uniformly, while we intuitively expect that in contrastive loss, misclassified nodes closer to the positive node should incur a lower penalty compared to those located farther away. However, conventional GCL approaches lack a mechanism to differentiate and appropriately penalize misclassified nodes based on proximity.

A straightforward approach for incorporating proximity information in the conventional GCL method can be computing a dense geodesic distance matrix for the entire graph or using spectral decompositions. However, these approaches can become expensive when applied in the context of contrastive learning. To tackle this problem, we introduce a **S**mooth **G**raph **C**ontrastive **L**earning (SGCL) method, which effectively integrates the geometric structure of graph views into a smoothed contrastive loss function. This loss function intuitively incorporates proximity information between nodes in positive and negative pairs through three developed smoothing approaches.

To extend the proposed contrastive loss for large-scale graphs, the GCL framework incorporates a mini-batch strategy. The integration of the mini-batch strategy significantly improves the efficiency of the model in handling large-scale graphs, which is a crucial requirement within the vanilla contrastive loss framework.

We evaluate the SGCL framework in node and graph classification tasks across benchmarks of various properties. Our results consistently demonstrate the superior performance of our method compared to state-of-the-art GCL methods. Our contributions are summarized as follows:

- We introduce a novel graph contrastive objective function, which effectively incorporates node proximity information to address uniform negative sampling limitations in conventional GCL methods.
- We introduce three formulations for smoothing in the contrastive learning objective, incorporating proximity information in the assignment of positive and negative pairs.
- We extend the proposed model for large-scale graphs by incorporating a mini-batch strategy into the GCL framework, enhancing model efficiency and computational scalability.
- We evaluate the GCL model for both node and graph classification on a diverse set of benchmarks with different scales and demonstrate its superior performance over state-of-the-art methods.

A comprehensive and detailed explanation of related work is presented in Appendix A.

## 2 BACKGROUND AND MOTIVATION

### 2.1 UNIFORM NEGATIVE SAMPLING

In the unsupervised GCL models introduced with two views $i$ and $j$, for each anchor node $v_t^{(i)}$ with feature embedding $\mathbf{h}_t^{(i)}$, the contrasting learning model defines a positive set $\mathcal{P}(v_t^{(i)}) = \{v_p^{(j)}\}_{p=1}^P$, consisting of $P$ elements, and a negative set $\mathcal{Q}(v_t^{(i)}) = \{v_q^{(j)}\}_{q=1}^Q$, comprising $Q$ elements.

In the absence of labeled information, these sets are confined to containing consistent samples within each graph view. Essentially, the positive set is formed by pairing embeddings in the two augmented graph views that align with the same node. Therefore, $P = 1$ and $\mathcal{P}(v_t^{(i)}) = \{v_p^{(j)}\}$, where $v_p^{(j)}$ in view $j$ corresponds to $v_t^{(i)}$ in view $i$. If corresponding nodes are in the same order in two views, then $t = p$. Additionally, all incongruent samples in each view $j$ are categorized as negative samples, $\mathcal{Q}(v_t^{(i)}) = \{v_q^{(j)}\}_{q=1,q\neq t}^{N-1}$.

Considering the ground truth, positive/negative pairs demonstrate semantic congruence/incongruence, particularly in relation to shared labels with the anchor. These pairs encompass samples affiliated with either the same class (positive) or different classes (negative). However, in the absence of labeled information, numerous incongruent nodes are false negative because they have the potential to be semantically similar to the anchor node but are instead categorized as negative pairs. This misalignment of the negative pairs adversely affects the learning process due to its inadvertent impact on the objective function. Consider the contrastive loss function 1, designed for each anchor node $v_t^{(i)}$. The objective is to minimize the distance between embeddings

of positive pair $\{v_t^{(i)}, v_t^{(i)}\}$ and simultaneously maximize the distance between embeddings of negative pairs $\{v_t^{(i)}, v_q^{(j)}\}_{q=1, q \neq t}^{N-1}$:

$$\mathcal{L}_{con}(v_t^{(i)}, V^{(j)}) = -\log \left( \frac{\exp\left(\mathbf{h}_t^{(i)}.\mathbf{h}_t^{(j)}/\tau\right)}{\exp\left(\mathbf{h}_t^{(i)}.\mathbf{h}_t^{(j)}/\tau\right) + \sum_{q=1, q \neq t}^{N-1} \exp\left(\mathbf{h}_t^{(i)}.\mathbf{h}_q^{(j)}/\tau\right)} \right) \tag{1}$$

Misalignment in negative pairs $\{v_t^{(i)}, v_k^{(j)}\}$ detrimentally impacts the learning process by introducing errors in the loss computation. The misalignment leads to an undesired increase in the loss, hindering the optimization process. Specifically, the GCL model increases the distance between misaligned negative pairs, and inadvertently separates semantically similar samples, leading to a degradation of overall performance.

Essentially, the negative pairs in the contrastive loss function are expected to contribute varying significance based on their proximity to the true positive node. However, in the conventional contrastive learning framework, which lacks information about the proximity of these nodes, all $N-1$ negative pairs are handled in a uniform manner. In other words, the conventional contrastive learning approach treats all misclassified nodes equally regardless of whether the misclassification occurs near the true positive or at a significant distance from it.

## 2.2 Extending graph geometry

The aforementioned limitations of conventional contrastive learning models arise from their incapacity to utilize semantic information during the training process. Nevertheless, there remains an advantage in exploiting the geometric information inherent in a graph to provide supplementary insights.

In conventional contrastive learning models, the positive pairs between two views are represented by a positive matrix $\Pi_{pos}^{(i,j)} \in \{0,1\}^{N \times N}$ with '1' on the diagonal and a negative matrix $\Pi_{neg}^{(i,j)} = 1 - \Pi_{pos}^{(i,j)} \in \{0,1\}^{N \times N}$ with '0' on the diagonal and '1' in the off-diagonal elements.

We propose a smoothing strategy that goes beyond simple binary categorization of matrices as positive or negative and applies a form of smoothing to the standard contrastive loss. This strategy allows nodes initially categorized as positive or negative to have values ranging from '0' to '1', indicating their degree of association with positive or negative samples, respectively. The smoothing process $\mathcal{S}(\Pi_{pos}^{(i,j)}, \mathbf{A}^{(i)})$ effectively integrates the geometric structure of a graph $\mathcal{G}^{(i)}$, enriching both positive and negative pairs by encompassing neighborhood relationships and capturing the broader context of the nodes $\mathcal{V}^{(i)}$.

Smoothing involves updating the values of the nodes iteratively based on the values of their neighboring nodes. This process helps to smooth the data while preserving the underlying graph structure. In a general setting, we take a binary positive matrix $\Pi_{pos}^{(i,j)}$ as the input of the smoothing approach $\mathcal{S}(\Pi_{pos}^{(i,j)}, \mathbf{A}^{(i)})$ to generate a smooth positive matrix $\tilde{\Pi}_{pos}^{(i,j)} \in [0,1]^{N \times N}$. The corresponding smoothed negative matrix $\tilde{\Pi}_{neg}^{(i,j)}$ is then obtained as $1 - \tilde{\Pi}_{pos}^{(i,j)}$. In the following, we formulate three existing formulations for prompting smoothing, including Taubin smoothing Taubin (2023), Bilateral smoothing Tomasi & Manduchi (1998), and Diffusion-based smoothing Gerig et al. (1992).

**Taubin smoothing** $\mathcal{S}_T(\mathbf{V}, \mathbf{L}, K, \mu, \tau)$ involves iteratively performing two stages of filtering utilized graph Laplacian matrix $\mathbf{L} \in \mathbb{R}^{N \times N}$ to smooth the binary matrix $\mathbf{V} \in \{0,1\}^{N \times D}$ as follows:

$$\mathbf{V}^{(k+1)} = (\mathbf{I} + \mu\mathbf{L})((\mathbf{I} + \tau\mathbf{L})\mathbf{V}^{(k)}) \tag{2}$$

The interior filter refers to the *positive Laplacian filter*, which operates by smoothing the values within the matrix $\mathbf{V}$ by moving each vertex to the average position of its neighbors. This process tends to blur the details, where $\tau > 0$ is a positive constant that controls the amount of smoothing. Subsequently, the *negative Laplacian filter* is employed to rectify the oversmoothing that occurred in the previous step. It moves each vertex in the opposite direction, preserving important geometric features. Here, $\mu(< -\tau)$ is a negative constant that corrects the oversmoothing from the previous step.

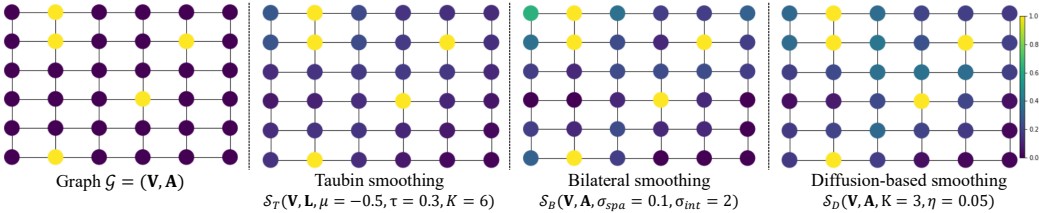

Figure 1: An illustrative example of the efficacy of the smoothing approaches on a grid graph $\mathcal{G}$. We color the grid according to the node value. In the left grid, initial values of 1 are represented in yellow, whereas nodes with zero values are depicted in dark purple. Each smoothing approach modifies the values of the zero nodes according to neighboring information.

**Bilateral smoothing** $\mathcal{S}_B(\mathbf{V}, \mathbf{A}, \sigma_{spa}, \sigma_{int})$ performs smoothing on a binary matrix $\mathbf{V}$ by combining information from nearby nodes, considering both spatial proximity and intensity similarity. Spatial proximity $d_{spa}$ captures the structural proximity between nodes within the graph by employing shortest path distance metrics. Intensity similarity $d_{int}$ is determined by evaluating the similarity in binary values between two nodes, typically quantified using metrics like the Hamming distance. The bilateral filter weight $w(i, j)$ is then computed, incorporating both criteria:

$$w(i,j) = \exp\left(-\frac{d_{spa}}{2\sigma_{spa}^2} - \frac{d_{int}}{2\sigma_{int}^2}\right), \tag{3}$$

where $\sigma_{spa}^2$ and $\sigma_{int}^2$ modulate the extent of both spatial and intensity smoothing, respectively.

The smoothed value of the central node $v_i$ is calculated as a weighted average of the binary values in row $i$ based on the calculated weights:

$$\tilde{\mathbf{v}}_i = \frac{\sum_{k=1}^{N} w(i,k)\mathbf{v}_k}{\sum_{k=1}^{N} w(i,k)} \tag{4}$$

**Diffusion-based smoothing** $\mathcal{S}_D(\mathbf{V}, \mathbf{A}, K, \eta)$ represents a method that employs diffusion equation to effectively diffuse information among nodes within a graph. It describes how data or attributes propagate from one node to its neighboring nodes over time, thereby achieving binary value smoothing. The smoothing process is first initialized with the original map matrix as the initial conditions, where each binary value serves as the starting "heat" at its respective node. Subsequently, the new value for each node is iteratively calculated based on the chosen diffusion equation and the binary values of its neighbors, considering the graph's connectivity and the diffusion process as $\mathbf{v}_i^{(k+1)} = \mathbf{v}_i^{(k)} + \eta \bar{\mathbf{v}}_i^{(k)}$, where $\bar{\mathbf{v}}_i^{(k)} = \sum_{j \in \mathcal{N}(v_i)} \mathbf{v}_j^{(k)}$ and a diffusion rate $\eta$ is applied to determine how much the binary value diffuses from one node to another.

It is worth noting that in the last step of all smoothing approaches, a masking step is applied to preserve the initial '1' values while ensuring that the other smoothed values remain unchanged.

Figure 1 illustrates an example of the efficacy of the smoothing approaches. We take a specific graph, such as a grid graph, and randomly establish a delta function, centered on specific vertices within this graph, resulting in the creation of a binary matrix. Subsequently, we employ a variety of smoothing techniques on this binary matrix. Given the uniform neighborhood structure of the grid, the resulting output exhibits a Gaussian-like distribution, which its center aligned to the initial vertex. However, the varied values in the smoothed matrix are indicative of the distinct strategies employed in the smoothing process.

In the context of contrastive learning on graphs, the positive matrix $\tilde{\Pi}_{pos}^{(i,j)}$ can be considered as a mapping from $\mathcal{G}^{(i)}$ to $\mathcal{G}^{(j)}$, with its rows and columns corresponding to nodes in $\mathcal{G}^{(i)}$ to $\mathcal{G}^{(j)}$, respectively. The goal of the smoothing approach is to extend this mapping to the neighbors of the paired nodes. In this specific context, since the rows of the positive matrix $\Pi pos^{(i,j)}$ are associated with nodes in $\mathcal{G}^{(i)}$, the smoothing approach utilizes the geometry of graph view $\mathcal{G}^{(i)}$. Similarly, for the positive matrix $\Pi_{pos}^{(j,i)}$, the smoothing approach utilizes to the geometric properties of the graph view $\mathcal{G}^{(j)}$.

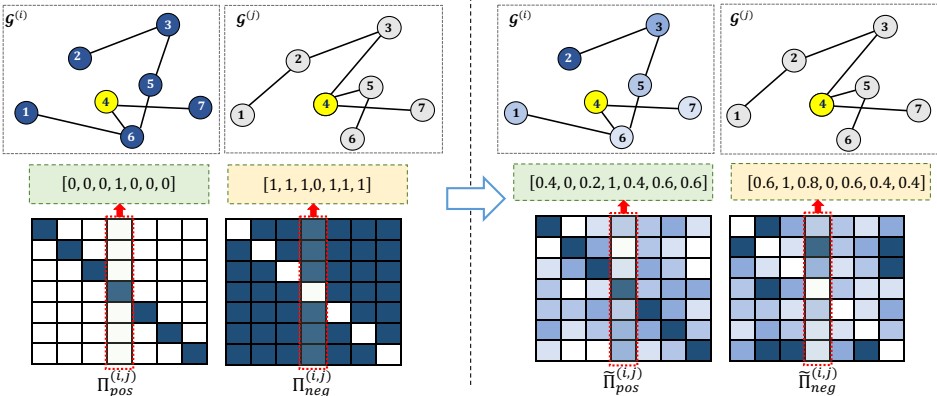

Figure 2: In the general context of conventional contrastive learning approaches, for every anchor node $v_4^{(i)}$ in $\mathcal{G}^{(i)}$, a corresponding positive node $v_4^{(j)}$ exists in $\mathcal{G}^{(j)}$, with all other node pairs being negative (left image). Smoothing techniques, which leverages the geometry of graph $\mathcal{G}^{(i)}$, effectively extract neighboring node information of an anchor node $v_4^{(i)}$ and generate smoothed positive and negative pairs matrices $\tilde{\Pi}_{pos}^{(i,j)}$ and $\tilde{\Pi}_{neg}^{(i,j)}$ (right image).

Figure 2 illustrates the distinctions between positive and negative pairs in the conventional contrastive learning framework and our proposed smoothed contrastive approach. Notably, when considering a specific anchor node $v_t^{(i)}$ in $\mathcal{G}^{(i)}$ paired with $v_k^{(j)}$ in $\mathcal{G}^{(j)}$, the graph information from $\mathcal{G}^{(i)}$ is employed to generate the smoothed positive and negative pairs matrices $\tilde{\Pi}_{pos}^{(i,j)}$ and $\tilde{\Pi}_{neg}^{(i,j)}$.

## 3   METHOD: SMOOTHED GRAPH CONTRASTIVE LEARNING

### 3.1   PRELIMINARIES

In the domain of unsupervised graph representation learning, we introduce an undirected graph $\mathcal{G} = (\mathcal{V}, \mathcal{E})$, where $\mathcal{V}$ constitutes the node set $\{v_1, v_2, ..., v_N\}$, and $\mathcal{E}$ denotes the edge set, formally captured as $\mathcal{E} \subseteq \mathcal{V} \times \mathcal{V}$. Within this contextual framework, we establish the definition of two pivotal matrices: the feature matrix $\mathbf{X} \in \mathbb{R}^{N \times F}$, wherein each $\mathbf{x}_i \in \mathbb{R}^F$ represents the feature vector associated with a distinct node $v_i$; and the adjacency matrix $\mathbf{A} \in \{0, 1\}^{N \times N}$, where the binary element $a_{i,j}$ equals 1 if and only if an edge exists between nodes $v_i$ and $v_j$.

The objective is to develop a GNN encoder $f_\theta(\mathbf{X}, \mathbf{A})$ that takes feature representations and graph structural characteristics of the graph as input and generates reduced-dimensional node embeddings $\mathbf{H} = f_\theta(\mathbf{X}, \mathbf{A}) \in \mathbb{R}^{N \times F'}$, where $F' \ll F$. Ultimately, the reduced-dimensional node embeddings prove to be invaluable assets in subsequent tasks, particularly in node classification.

### 3.2   FRAMEWORK

The main objective of the proposed framework is unsupervised node classification, with a specific focus on effectively addressing medium and large-scale graphs. We introduce a novel framework, termed **S**moothed **G**raph **C**ontrastive **L**earning (SGCL), designed to construct node embeddings by seamlessly incorporating the geometric structure of augmented graphs to facilitate a smooth alignment between positive and negative pairs.

The comprehensive architecture of our framework is visually illustrated in Figure 3. In the following sections, we will outline the sequential processing steps of the proposed framework.

#### 3.2.1   SUBGRAPH GENERATING

In order to address the challenge of scalability and to accommodate the contrastive loss for large-scale graphs, we leverage the random-walk mini-batches generation approach, proposed by Graph-

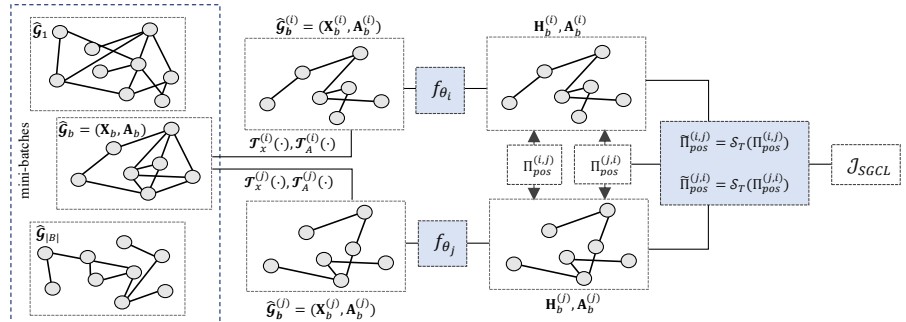

Figure 3: Overview of the Proposed SGCL Model. The SGCL model utilizes a mini-batch generation approach to generate $|B|$ subgraphs (mini-batches). For each subgraph $\hat{\mathcal{G}}_b$, two distinct augmentation methods are employed to produce two different views $\hat{\mathcal{G}}_b^{(i)}$ and $\hat{\mathcal{G}}_b^{(j)}$. The GCN encoders $f_{\theta_i}$ and $f_{\theta_j}$ are then utilized to learn feature embeddings $\mathbf{H}_b^{(i)}$ and $\mathbf{H}_b^{(j)}$, respectively. Subsequently, two smoothed positive matrices $\tilde{\Pi}_{pos}^{(i,j)}$ and $\tilde{\Pi}_{pos}^{(j,i)}$ are computed by leveraging the geometric structures of the graphs. These matrices are used in the contrastive loss $\mathcal{J}_{SGCL}$ to encourage the smooth interaction of node representations between the two views. This loss function, scores the agreement between these representations for all batches, serving as training loss.

SAINT Zeng et al. (2020), to generate subgraphs from a given graph. More specifically, an entire graph $\mathcal{G}$ is partitioned into a set of $|B|$ mini-batches denoted as $\hat{\mathcal{G}} = \{\hat{\mathcal{G}}_1, \ldots, \hat{\mathcal{G}}b, \ldots, \hat{\mathcal{G}}_{|B|}\}$, where each $\hat{\mathcal{G}}_b = (\hat{\mathcal{V}}_b, \hat{\mathcal{E}}_b)$ represents a sampled subgraph. It is essential to note that the construction of subgraphs varies depending on the specific sampling approach employed. Leveraging the insights gained from the variance analysis within GraphSAINT, it introduces a collection of lightweight and efficient mini-batch generation approaches, which are detailed in Appendix B.

### 3.2.2 GENERATING GRAPH VIEWS VIA AUGMENTATION

Two categories of graph augmentation approaches are broadly used in GCL models: (1) feature-based augmentations, which manipulate the initial node features, often through operations such as masking or the introduction of Gaussian noise, and (2) structure-based augmentations and corruptions, which involve alterations to the underlying structure of graphs, such as adding or removing connectivities and sub-sampling. In the proposed framework, such strategies as sub-sampling augmentation can pose challenges, particularly in constructing positive and negative node pairs across multiple views. Empirical evidence consistently demonstrates the effectiveness of two essential augmentation strategies for optimal results: edge dropping and node feature masking. Their combination generates two distinct graph views for each mini-batch $\hat{\mathcal{G}}_b$, referred to as $\hat{\mathcal{G}}_b^{(1)}$ and $\hat{\mathcal{G}}_b^{(2)}$.

More specifically, in each view $i$, we construct the augmented graph $\hat{\mathcal{G}}_b^{(i)}$ as follows: $\hat{\mathcal{G}}_b^{(i)} = (\mathcal{T}_x^{(i)}(\mathbf{X}_b), \mathcal{T}_A^{(i)}(\mathbf{A}_b))$, where $\mathcal{T}_x(\mathbf{X}) = \mathbf{X} \odot (1 - M_X)$ and $\mathcal{T}_A(\mathbf{A}) = \mathbf{A} \odot (1 - M_A) + (1 - \mathbf{A}) \odot M_A$. The feature value mask matrix $M_X \sim \mathcal{N}(0, \Sigma)$ is employed to replace original values with Gaussian noise. Additionally, the perturbation location mask $M_A$ uses a Bernoulli distribution and randomly drops edges from the adjacency matrix with a specified probability.

### 3.2.3 ENCODERS

The main component of the proposed framework is the encoder network with a parameter set $\theta$, denoted as $f_\theta$. This network operates on an augmented graph as input, generating reduced-dimensional feature embeddings for each node within the graph. Subsequently, these reduced-dimensional node embeddings play a pivotal role in various follow-up tasks, with a particular emphasis on node classification. We have the flexibility to employ any encoders capable of constructing node embeddings without imposing constraints. Within this framework, we opt for the widely adopted Graph Convolution Network (GCN) Kipf & Welling (2017) as the foundational graph encoders. For each view $i$,

we employ a dedicated graph encoder $\mathbf{H} = f_{\theta_i}(\mathbf{X}, \mathbf{A}) : \mathbb{R}^{N \times F} \times \mathbb{R}^{N \times N} \to \mathbb{R}^{N \times F'}$ that leverages adjacency and feature matrices as two congruent structural perspectives of GCN layers [1].

The GCN operates across multiple layers, wherein the message-passing process is recurrently applied at each layer. The node representations are updated in a layer-wise manner: $\mathbf{H}^{(l+1)} = \sigma\left(\tilde{\mathbf{D}}^{-1/2}\tilde{\mathbf{A}}\tilde{\mathbf{D}}^{-1/2}\mathbf{H}^{(l)}\mathbf{W}^{(l)}\right)$, where $\tilde{\mathbf{A}}$ denotes the symmetrically normalized adjacency matrix, calculated as $\tilde{\mathbf{A}} = \mathbf{A} + \mathbf{I}$ with diagonal matrix $\mathbf{I} \in \mathbb{R}^{N \times N}$, $\tilde{\mathbf{D}}_{ii} = \sum_j \tilde{\mathbf{A}}_{ij} \in \mathbb{R}^{N \times N}$ is the degree matrix, $\mathbf{W}^{(l)} \in \mathbb{R}^{F_l \times F_{l+1}}$ is the learned weight matrix for layer $l$, $\sigma$ is activation function, and $\mathbf{H}^{(l)} \in \mathbb{R}^{N \times F_l}$ is the node representation in layer $l$.

### 3.2.4 SMOOTHED CONTRASTIVE OBJECTIVE

To end-to-end training of the encoders and promote node representations, we introduce an innovative contrastive objective. This objective utilizes a smoothed positive pairs matrix $\tilde{\Pi}_{pos}^{(i,j)}$ to encourage the agreement between encoded embeddings of two nodes, namely, $v_t^{(i)}$ and $v_p^{(j)}$, in two different views with degree $\hat{\pi}_{pos}^{(i,j)}(t,p)$, while also distinguish their embeddings with a degree of $1 - \hat{\pi}_{pos}^{(i,j)}(t,p)$. The objective is defined as follows:

$$\mathcal{L}_{SGCL}^{(i,j)} = \| \tilde{\Pi}_{pos}^{(i,j)} \odot (1 - \mathbf{C}^{(i,j)}) \|_F^2 + \lambda \| (1 - \tilde{\Pi}_{pos}^{(i,j)}) \odot \mathbf{C}^{(i,j)} \|_F^2 \qquad (5)$$

where $\lambda > 0$ defines the trade-off between two terms during optimizing and $\mathbf{C}^{(i,j)}$ is the similarity matrix between the normalized embeddings $\hat{\mathbf{H}}^{(i)}$ and $\hat{\mathbf{H}}^{(j)}$ of identical networks, computed via cosine similarity along the "feature" dimension:

$$\mathbf{C}^{(i,j)} = \frac{\hat{\mathbf{H}}^{(i)}\hat{\mathbf{H}}^{(j)^T}}{\| \hat{\mathbf{H}}^{(i)} \| \| \hat{\mathbf{H}}^{(j)} \|} \qquad (6)$$

In the proposed contrastive objective, the first term enforces the stability of the preservation in the embeddings of positive pairs by minimizing the discrepancy between 1 and each element of $\mathbf{C}^{(i,j)}$. This alignment is achieved with the values in the smoothed positive pairs matrix $\tilde{\Pi}_{pos}^{(i,j)}$, effectively equivalent to maximizing $\mathbf{C}^{(i,j)}$ for positive pairs. Conversely, the second term actively promotes a substantial diversity in the embeddings of negative pairs by minimizing each element of $\mathbf{C}^{(i,j)}$ concerning the values in the smoothed negative pairs matrix $\tilde{\Pi}_{neg}^{(i,j)} = 1 - \tilde{\Pi}_{pos}^{(i,j)}$.

At each training epoch, the smoothed positive pairs matrix $\tilde{\Pi}_{pos}^{(i,j)}$ is computed by incorporating the geometric structure of the graph view $\mathcal{G}^{(i)}$ into each smoothing approaches mentioned in Section 2.2. For instance, we use Taubin smoothing, which yields $\tilde{\Pi}_{pos}^{(i,j)} = \mathcal{S}_T(\Pi_{pos}^{(i,j)}, \mathbf{L}^{(i)}, \mu, \tau)$ and $\tilde{\Pi}_{pos}^{(j,i)} = \mathcal{S}_T(\Pi_{pos}^{(j,i)}, \mathbf{L}^{(j)}, \mu, \tau)$. Ultimately, we optimize the model parameters by considering all $|B|$ batches within the given graph concerning the overall innovated contrastive loss $\mathcal{J}_{SGCL} = \frac{1}{2|B|} \sum_{b=1}^{|B|} (\mathcal{L}_{SGCL}^{(i,j)} + \mathcal{L}_{SGCL}^{(j,i)})$.

An ablation study evaluating the impact of different terms in the contrastive objective, along with an exploration of the hyperparameter $\lambda$, is presented in Appendix F.2.

## 4 EXPERIMENTS

We conduct empirical evaluations of our proposed SGCL model through node and graph classification tasks, using a variety of publicly available benchmark datasets. For node classification, the benchmarks encompass a wide range of graph sizes, including smaller to medium-scaled ones such as Cora, Citeseer, Pubmed Sen et al. (2008), CoauthorCs Sinha et al. (2015), Computers, and Photos McAuley et al. (2015), as well as larger datasets like ogbn-arxiv, ogbn-products, ogbn-proteins, and all of which are sourced from the Open Graph Benchmark Hu et al. (2020). For graph classification, we employ MUTAG Kriege & Mutzel (2012), PTC Kriege & Mutzel (2012), IMDB-Binary Yanardag & Vishwanathan (2015), PROTEINS Wale et al. (2008), and ENZYMES Borgwardt et al. (2005) benchmarks. Appendix C provides comprehensive details of the benchmarks.

---

[1] For the sake of simplicity, we omit the view index in superscript and the batch index in subscript.

## 4.1 BASELINES

In our empirical study, we incorporate a variety of models for comparison. For node classification, these models encompass representative node classification models, as well as recently-introduced graph contrastive learning models, such as DGI Veličković et al. (2019), GRACE Zhu et al. (2020), MVGRL Hassani & Khasahmadi (2020), GBT Bielak et al. (2022), BGRL Thakoor et al. (2022), CGRA Duan et al. (2023), and GRLC Peng et al. (2023) serving as our baseline models. For graph classification, we employ seven state-of-the-art methods for graph contrastive learning, including InfoGraph Sun et al. (2019), GraphCL You et al. (2020), MVGRL Hassani & Khasahmadi (2020), BGRL Thakoor et al. (2022), AD-GCL Suresh et al. (2021), LaGraph Xie et al. (2022), and CGRA Duan et al. (2023). Supplementary details of hyperparameters and architecture are described in Appendix D. The computational cost of the proposed model, in comparison to the baselines, is also provided in Appendix E.2.

## 4.2 NODE CLASSIFICATION

Node classification is one of the important downstream tasks, employed to reflect the effectiveness of the learned graph representation. In the first experiment, we employ six small and medium-scale benchmark datasets: Cora, Citeseer, Pubmed, CoauthorCS, Computers, and Photo. The proposed models are derived by incorporating three distinct smoothing techniques in the proposed models: SGCL-T (Taubin smoothing), SGCL-B (Bilateral smoothing), and SGCL-D (Diffusion-based smoothing). Table 1 reports the performance of the proposed SGCL model and compares the results with baseline models. To generate mini-batches in this experiment, we utilize a random-walk sampling, as outlined in Appendix B. A summary of the results derived from other mini-batching approaches is reported in Table 8.

Table 1: Comparison of node classification accuracies of proposed models vs. baselines on small and medium-scaled graphs (mean ± std).

| Model | Cora | Citeseer | Pubmed | CoauthorCS | Computers | Photo |
|---|---|---|---|---|---|---|
| DGI Veličković et al. (2019) | 82.43±1.3 | 70.96±2.4 | 83.13±0.6 | 91.67±0.7 | 65.07±1.2 | 77.15±1.1 |
| GRACE Zhu et al. (2020) | 83.16±1.9 | 68.68±1.3 | 85.68±0.3 | 91.43±0.7 | 81.89±0.6 | 89.33±1.7 |
| MVGRL Hassani & Khasahmadi (2020) | 85.44±1.8 | 72.51±2.9 | 86.33±0.7 | 92.91±0.5 | 85.90±0.6 | 91.48±0.9 |
| BGRL Thakoor et al. (2022) | 79.12±2.7 | 63.77±1.7 | 84.17±0.7 | 91.6±0.65 | 84.27±0.5 | 92.31±0.4 |
| GBT Bielak et al. (2022) | 76.91±1.8 | 59.46±2.9 | 85.92±0.5 | 91.45±0.8 | **88.11±0.9** | 91.29±0.7 |
| SGCL-T | 86.54±1.4 | **73.71±2.0** | **86.57±0.5** | 92.99±0.4 | 87.83±1.4 | **93.05±0.9** |
| SGCL-B | **87.50±1.7** | 73.65±1.3 | 86.13±0.6 | 93.15±0.4 | 87.89±0.7 | 92.31±1.3 |
| SGCL-D | 85.22±0.8 | 72.66±2.4 | 86.27±0.5 | **93.21±0.4** | 85.47±1.2 | 91.86±0.7 |

The results indicate that our proposed model outperforms the majority of benchmarks, providing validation for the effectiveness of our proposed learning framework. In comparison to other benchmarks, the "Computers" graph exhibits a notably high average node degree but a lower degree of homophily (see Table 4). Consequently, this reduces the significance of neighboring nodes in the proposed smoothing approaches, leading to performance degradation compared to MVGRL.

For further investigation and to facilitate a comprehensive comparison with the existing state-of-the-art, we replicated this experiment in a full-batch scenario, adhering to the commonly employed data split in self-supervised learning as provided in the Open Graph Benchmark [2], as described in Appendix E.1 (Table 6).

The observed performance verifies the enhanced capacity achieved through the utilization of the geometric structure inherent in graphs, enabling improved exploration of positive and negative pairs within the conventional contrastive learning framework. It is worth noting that SGCL-T, which leverages Laplacian information using the Tubin smoothing approach, provides better and more promising results, even slightly outperforming the other smoothing approaches.

In the second experiment, we perform experiments on three large-scale graphs: ogbn-arxiv, ogbn-products, and ogbn-proteins. In this experiment, the significance of the mini-batch generation step of the proposed framework becomes more prominent since employing full-batch large-scale graphs can impose considerable demands on GPU memory due to the necessity of loading all node embeddings

---

[2] https://ogb.stanford.edu/docs/nodeprop/

onto the GPU. In this experiment, we employ a random-walk sampling approach to generate mini-batches. The results presented in Table 2, demonstrate that the proposed SGCL model consistently outperforms other contrastive learning methods on large-scale graphs.

Table 2: Comparison of node classification accuracies of proposed models vs. baselines on large-scaled graphs (mean ± std).

| Model | ogbn-arxiv | ogbn-products | ogbn-proteins |
|---|---|---|---|
| DGI Veličković et al. (2019) | 67.07±0.5 | 68.68±0.6 | 94.11±0.1 |
| GRACE Zhu et al. (2020) | 67.92±0.4 | 72.10±0.7 | 94.11±0.2 |
| MVGRL Hassani & Khasahmadi (2020) | 60.68±0.5 | 69.90±0.9 | 93.87±0.3 |
| BGRL Thakoor et al. (2022) | 63.88±0.2 | 66.23±0.5 | 92.94±0.3 |
| GBT Bielak et al. (2022) | 69.05±0.3 | 65.74±0.4 | 94.07±0.3 |
| SGCL-T | **69.30±0.5** | **75.97±0.1** | **94.64±0.2** |
| SGCL-B | 69.24±0.3 | 74.33±0.4 | 93.55±0.2 |
| SGCL-D | 69.03±0.4 | 74.15±0.2 | 93.19±0.1 |

It's worth noting that ogbn-products serves as an excellent benchmark for our proposed models due to two key advantages. Firstly, its high homophily rate increases the likelihood of discovering neighboring nodes of positive pairs as new positive pairs, enhancing the performance of the model. Secondly, by using mini-batch graphs, instead of the full-batch graph with numerous connected components, we can effectively move beyond the extremely small components. This approach provides richer neighboring information, resulting in the generating of more efficient augmented graphs that contribute to the improved performance of the contrastive loss framework.

### 4.3 GRAPH CLASSIFICATION

Graph classification is another important downstream task, employed to reflect the effectiveness of the learned graph representation. In this experiment, we follow the InfoGraph Sun et al. (2019) setting for graph classification and compare the accuracy with self-supervised state-of-the-art methods. The results reported in Table 3 indicate that, in comparison to the best-performing state-of-the-art methods, the proposed model demonstrates enhanced accuracy for IMDB-BINARY, PROTEINS, and ENZYMES, while maintaining comparable accuracy on other benchmarks. It's worth mentioning that the accuracies of all models are reported from their respective published papers, except for the BGRL results, which we reproduced under the same experimental setting.

Table 3: Comparison of graph classification accuracies of proposed models vs. baselines (mean ± std).

| Model | IMDB-Binary | PTC | MUTAG | PROTEINS | ENZYMES |
|---|---|---|---|---|---|
| InfoGraph Sun et al. (2019) | 73.0±0.9 | 61.7±1.4 | 89.0±1.1 | 74.4±0.3 | 50.2±1.4 |
| GraphCL You et al. (2020) | 71.1±0.4 | 63.6±1.8 | 86.8±1.3 | 74.4±0.5 | 55.1±1.6 |
| MVGRL Hassani & Khasahmadi (2020) | 74.2±0.7 | 62.5±1.7 | 89.7±1.1 | 71.5±0.3 | 48.3±1.2 |
| AD-GCL Suresh et al. (2021) | 71.5±1.0 | 61.2±1.4 | 86.8±1.3 | 75.0±0.5 | 42.6±1.1 |
| BGRL Thakoor et al. (2022) | 72.8±0.5 | 57.4±0.9 | 86.0±1.8 | 77.4±2.4 | 50.7±9.0 |
| LaGraph Xie et al. (2022) | 73.7±0.9 | 60.8±1.1 | 90.2±1.1 | 75.2±0.4 | 40.9±1.7 |
| CGRA Duan et al. (2023) | 75.6±0.5 | **65.7±1.8** | **91.1±2.5** | 76.2±0.6 | 61.1±0.9 |
| SGCL-T | 75.2±2.8 | 64.0±1.6 | 89.0±2.3 | 79.4±1.9 | **65.3±3.6** |
| SGCL-B | 73.2±3.7 | 62.5±1.8 | 87.0±2.8 | **81.6±2.3** | 63.7±1.6 |
| SGCL-D | **75.8±1.9** | 62.6±1.4 | 86.0±2.6 | 81.5±2.3 | 64.3±2.2 |

## 5 CONCLUSION

Conventional Graph Contrastive Learning (GCL) methods use a straightforward approach for distinguishing positive and negative pairs, often leading to challenges in uniformly identifying negative pairs regardless of their proximity. In this paper, we introduced a Smooth Graph Contrastive Learning (SGCL) method, which incorporates the geometric structure of graph views into a smoothed contrastive loss function. SGCL offers an intuitive way that employs three smoothing approaches to consider proximity information when assigning positive and negative pairs. The GCL framework is enhanced for large-scale graphs by incorporating a mini-batch strategy, leading to improved model efficiency and computational scalability. The evaluations, conducted on graphs of varying scales, consistently show that SGCL outperforms state-of-the-art GCL approaches in node and graph classification tasks. This emphasizes the effectiveness of the smoothed contrastive loss function in capturing and utilizing proximity information, ultimately improving the performance of the SGCL.

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

## A   RELATED WORK

### A.1   GRAPH REPRESENTATION LEARNING

In recent years, graph neural networks (GNNs) have made significant progress, by the emergence of a multitude of methods dedicated to enhancing graph representation learning. These methods have been designed to address various aspects of network embeddings, including proximity, structure, attributes, learning paradigms, and scalability Wu et al. (2021); Bronstein et al. (2017). Among the notable GNN approaches, Graph Convolutional Networks (GCN) Kipf & Welling (2017) is one of the foundational GNNs that uses convolutional operations to capture local and global information from neighboring nodes, making them effective for tasks like node classification. To overcome the constraints associated with conventional graph convolutions and their approximations, the Graph Attention Network (GAT) Veličković et al. (2018) introduces the notion of masked self-attentional layers, thereby enhancing its capacity to capture crucial node relationships. By integrating an autoregressive moving average (ARMA) filter, GNN-ARMA Bianchi et al. (2021) extends the functionality of GNNs to adeptly capture global graph structures. GWCN, as proposed in Xu et al. (2019a); Behmanesh et al. (2022), utilizes graph wavelets as spectral bases for convolution. This innovative approach enables the modeling of both local and global structural patterns within graphs. GRAND Chamberlain et al. (2021) presents an interesting perspective on graph convolution networks (GCNs) by interpreting them as a solution to the heat diffusion equation. TIDE Behmanesh et al. (2023) introduces an innovative approach to tackle the oversmoothing challenge in the message-passing-based approaches by leveraging the diffusion equation to enable efficient and accurate long-distance communication between nodes in a graph.

However, it's essential to emphasize that the majority of these methods depend on supervised data, and this can be a significant limitation in real-world applications due to the difficulties associated with acquiring labeled datasets. Several traditional unsupervised graph representation learning methods are designed to learn meaningful representations of nodes in a graph without the need for labeled data or explicit supervision. DeepWalk Perozzi et al. (2014) employs random walks and skip-gram modeling to capture local graph structure, while node2vec Grover & Leskovec (2016) extends this approach with a versatile biased random walk strategy encompassing breadth-first and depth-first exploration. LINETang et al. (2015) focuses on preserving both first-order and second-

order proximity information in large-scale networks, and GraphSAGE Hamilton et al. (2017) combines random walk sampling and aggregation to capture both local and global graph structure. HOPE Ou et al. (2016) leverages higher-order proximity information to capture structural patterns beyond pairwise node relationships in graphs.

## A.2  GRAPH CONTRASTIVE LEARNING

Self-supervised learning (SSL) has emerged as a powerful paradigm for mitigating the challenges posed by expensive, limited, and imbalanced labels. It enables deep learning models to train on unlabeled data, reducing the reliance on annotated labels Xie et al. (2023).

Contrastive Learning (CL) is a popular SSL technique known for its simplicity and strong empirical performance. Its fundamental objective is to create meaningful representations by pushing dissimilar pairs apart and pulling similar pairs closer together. Graph Contrastive Learning (GCL) extends the concept of CL to the domain of graphs. However, dealing with the irregular structure of graph data presents more complex challenges in designing strategies for constructing positive and negative samples compared to CL applied to visual or natural language data Wu et al. (2023).

Numerous papers have emerged to address the challenges associated with GCL. These papers primarily focus on sharing valuable insights and practical approaches for three key elements of contrastive learning: data augmentation, pretext tasks, and contrastive objective Wu et al. (2023).

Deep Graph Infomax (DGI) Veličković et al. (2019) and InfoGraph Sun et al. (2019) are two fundamental contrastive learning models that train a node encoder by maximizing mutual information between the node representation and the global graph representation. DGI is designed for node representation learning, whereas InfoGraph focuses on graph-level representations.

MVGRL Hassani & Khasahmadi (2020) is one of the recent GCL approaches that accomplishes the learning of both node and graph-level representations by considering two matrices, namely adjacency and diffusion, as congruent views of a standard contrastive framework.

The fundamental of the aforementioned GCL approaches is the maximization of local-global mutual information within a framework. However, they all rely on a readout function to generate the global graph embedding which this function can be overly restrictive and may not always be achievable. Moreover, for approaches like DGI Veličković et al. (2019), there is no guarantee that the resulting graph embedding can effectively capture valuable information from the nodes, as it may not adequately preserve the distinctive features found in node-level embeddings.

Several GCL approaches, including GRACE Zhu et al. (2020), GraphCL You et al. (2020), and CSSL Zeng & Xie (2021), deviate from the conventional approach of contrasting local-global mutual information. Notably, these methods do not rely on making assumptions about the use of injective readout functions to generate the graph embedding.

The effectiveness of the GCL models depends on comparing each item with many negative points He et al. (2020). However, relying on these negative examples is problematic, especially for graphs, where defining negative samples in a meaningful manner is particularly difficult.

To address this challenge, several Graph contrastive learning methods have emerged that eliminate the need for explicit negative pairs. For instance, BGRL applies the BYOL method Grill et al. (2020) to graphs as a GCL approach that doesn't depend on negative pairs Thakoor et al. (2022). Additionally, Graph Barlow Twins (GBT) utilizes a cross-correlation-based loss function instead of negative samples Bielak et al. (2022).

In our approach, we neither treat negative pairs the same way as in GRACE nor ignore them like in BGRL. Instead, we make use of negative pairs within the contrastive loss, but with a unique approach – we use the geometric structure of graphs to effectively consider proximity among negative pairs in contrastive learning, rather than treating them all the same.

## B  MINI-BATCH GENERATING APPROACHES

**Random node sampler** approach randomly selects a subset of nodes from a given graph $\mathcal{G} = (\mathcal{V}, \mathcal{E})$ according to a probability distribution $P(v)$, where $v$ represents individual nodes in the graph. The

distribution $P(v)$ assigns a probability to each node, indicating the likelihood of that node being included in the sampled subset $\mathbb{V}_s$.

**Random edge sampler** approach randomly selects edges from a given graph $\mathcal{G} = (\mathcal{V}, \mathcal{E})$ based on a predefined probability distribution. For each edge $e$ in the set of edges $\mathcal{E}$, an independent decision is made to determine whether it should be included in the subgraph $\mathcal{G}_s$. This decision is guided by a probability value $P(e)$ assigned to each edge. The sampler incorporates a budget parameter $m$ to constrain the expected number of sampled edges, ensuring that $\sum P(e) = m$, as described in Zeng et al. (2020).

**Random walk sampler** approach begins by randomly selecting $r$ root nodes as starting points on the entire graph $\mathcal{G} = (\mathcal{V}, \mathcal{E})$. From each of these starting nodes, random walks of length $r$ are conducted to generate subgraphs Zeng et al. (2020). To manage the potential issue of generating excessively large subgraphs, a batch size parameter $m$ is commonly employed, ensuring the approximate number of samples per batch.

**Ego graph sampler** approach generates subgraphs centered around a specific "ego" node in a graph $\mathcal{G} = (\mathcal{V}, \mathcal{E})$. This mini-batch generation approach provides a localized perspective on the graph by constructing a $k$-hop ego-graph centered at node $v_i$, where "$k$-hop" indicates that the subgraph includes nodes that can be reached within $k$ steps from $v_i$. Importantly, the sampler ensures that the maximum distance between $v_i$ and any other nodes within the ego-graph is limited to $k$, as expressed mathematically by $\forall v_j \in \mathcal{V}, \mid d(v_i, v_j) \mid < k$ Zhu et al. (2021a).

## C  PROPERTIES AND STATISTICS OF THE BENCHMARKS

The properties of different graph datasets used in the node and graph classification experiments are provided in Table 4 and 5, respectively. The homophily rate $h$ denotes the degree to which nodes in the graph connect with similar nodes (homophily) versus nodes with dissimilar nodes (heterophily). The diameter of large-scaled graphs is performed using Breadth-First Search (BFS) from a sample of 1,000 nodes selected at random.

Table 4: The statistics of the datasets for node classification evaluation

| Scale | Dataset | #Nodes | #Edges | #Feature | #Class | #CC | h% | Avg. N.D. | Diameter |
|-------|---------|--------|--------|----------|--------|-----|-----|-----------|----------|
| Small | Cora | 2,485 | 5,069 | 1,433 | 7 | 78 | 80.4 | 4.08 | 19 |
| | Citeseer | 2,120 | 3,679 | 3,703 | 6 | 438 | 73.5 | 3.47 | 28 |
| Medium | PubMed | 19,717 | 44,324 | 500 | 3 | 1 | 80.2 | 4.5 | 18 |
| | CoauthorCs | 18,333 | 81,894 | 6,805 | 15 | 1 | 80 | 8.93 | 24 |
| | Computers | 13,381 | 245,778 | 767 | 10 | 314 | 77.7 | 36.74 | 10 |
| | Photos | 7,487 | 119,043 | 745 | 8 | 136 | 82.7 | 31.8 | 11 |
| Large | ogbn-arxiv | 169,343 | 1,166,243 | 128 | 40 | 1 | 65.4 | 13.67 | 23 |
| | ogbn-products | 2,449,029 | 61,859,140 | 100 | 47 | 52,658 | 80.8 | 51.54 | 27 |
| | ogbn-proteins | 132,534 | 39,561,252 | 8 | 94 | 1 | 91 | 597 | 9 |

**#CC**: Number of connected components, **h%**: Homophily rate, **Avg. N.D**: Average node degrees

Table 5: The statistics of the datasets for graph classification evaluation

| Dataset | #Graph | Avg. node | Avg. edge | #Features | #Class |
|---------|--------|-----------|-----------|-----------|--------|
| MUTAG | 188 | 17.9 | 39.6 | 7 | 2 |
| PTC | 344 | 14.29 | 14.69 | 19 | 2 |
| IMDB-Binary | 1,000 | 19.8 | 193.1 | 1 | 2 |
| PROTEINS | 1,113 | 39.1 | 145.6 | 3 | 2 |
| ENZYMES | 600 | 32.63 | 124.3 | 3 | 6 |

## D  EXPERIMENTAL SETUP

In all experiments, we follow the linear evaluation scheme outlined in Veličković et al. (2019). Initially, we start by training the '2-layer' GCN encoders using the proposed SGCL framework in an unsupervised manner. The training process consists of 200 iterations, and we utilize the Adam optimizer with a learning rate of $1e-3$. Subsequently, the obtained embeddings are used to perform

node or graph classification on a downstream task, employing a $l_2-$ regularized logistic regression classifier. The mean classification accuracy, along with the standard deviation, is then reported on the test nodes after conducting 5000 training runs.

In the mini-batch scenario of the node classification task, we employ a random-walk batch generation approach to create subgraphs from the input graph. We then conduct node classification for a downstream task, with a split ratio of 0.1/0.1/0.8 for train/validation/test. In both graph classification and the full-batch scenario of node classification, we follow the widely adopted data split used in self-supervised learning, as outlined in the Open Graph Benchmark.

To implement the proposed model, we leveraged the extensive capabilities offered by the PyGCL library, as introduced in Zhu et al. (2021b). For the graph augmentation, we employ the augmentor base class provided by PyGCL, which includes Edge Removing (ER) and Node Feature Masking (FM), both with a drop probability of 0.5.

Additionally, for a comprehensive comparison, we conducted experiments on all baselines using the PyGCL library since there was a lack of extensive baseline experimentation, especially for large-scale graphs. The implementation of all experiments will be made available after the acceptance of this draft.

In the mini-batch generation using the random-walk sampler, we set the batch size to 2000 with a random walk length of 4 and 3 starting root nodes for all benchmark datasets. However, for the ogbn-products benchmark, we use a batch size of 500 with a random walk length of 20.

In the smoothing techniques, we set the parameters as follows: For Taubin smoothing, we use $\mu = -0.4$, $\tau = 0.3$, and $K = 2$. In the case of Bilateral smoothing, we employ $\sigma_{spa} = 0.1$ and $\sigma_{init} = 2$. Finally, for Diffusion-based smoothing, we utilize $\eta = 0.03$ and $K = 2$.

All experiments are implemented using PyTorch 1.13.1 and PyTorch Geometric 2.2.0 and conducted on NVIDIA A100 GPUs with 40GB of memory.

# E  SUPPLEMENTARY EXPERIMENTS

## E.1  NODE CLASSIFICATION IN FULL-BATCH SCENARIO

Given the computational resource constraints, we are able to conduct experiments on small and medium-scale benchmarks using a full-batch scenario. As a result, to establish a fair and robust comparison with the existing state-of-the-art methods, we replicated the node classification experiment in a full-batch scenario, adhering to the commonly employed data split in self-supervised learning as provided in the Open Graph Benchmark. Table 6 demonstrates that, without any mini-batch generating step, SGCL outperforms state-of-the-art methods in four out of six benchmarks. However, it's worth noting that the superiority of SGCL with the mini-batch generating step is more pronounced.

Table 6: Comparison of node classification accuracies of proposed models vs. baselines on small and medium-scaled graphs in *full-batch* scenario (mean ± std).

| Model | Cora | Citeseer | Pubmed | CoauthorCS | Computers | Photo |
|---|---|---|---|---|---|---|
| DGI Veličković et al. (2019) | 76.28±0.04 | 69.33±0.14 | 83.79±0.08 | 91.63±0.08 | 71.96±0.06 | 75.27±0.02 |
| GRACE Zhu et al. (2020) | 81.80±0.19 | 71.35±0.07 | 85.86±0.05 | 91.57±0.14 | 84.77±0.06 | 89.50±0.06 |
| MVGRL Hassani & Khasahmadi (2020) | 84.98±0.11 | 71.29±0.04 | 85.22±0.04 | 91.65±0.02 | 88.55±0.02 | 91.90±0.08 |
| BGRL Thakoor et al. (2022) | 80.21±1.14 | 66.33±2.10 | 81.78±1.06 | 90.19±0.82 | 84.24±1.32 | 89.56±1.01 |
| GBT Bielak et al. (2022) | 79.32±0.31 | 65.78±1.33 | **86.35±0.48** | 91.87±0.07 | **90.43±0.18** | 92.23±0.18 |
| CGRA Duan et al. (2023) | 82.71±0.01 | 69.23±1.19 | 82.15±0.46 | 91.26±0.27 | 89.76±0.36 | 91.54±1.06 |
| GRLC Peng et al. (2023) | 83.50±0.24 | 70.02±0.16 | 81.20±0.20 | 90.36±0.27 | 88.54±0.23 | 91.80±0.77 |
| SGCL-T | 84.45±0.04 | 71.26±0.06 | 84.11±0.08 | 92.14±0.09 | 86.81±0.01 | **92.71±0.05** |
| SGCL-B | **85.08±0.12** | **72.77±0.33** | 83.67±0.06 | **92.16±0.15** | 88.24±0.05 | 92.43±0.03 |
| SGCL-D | 84.47±0.25 | 70.32±0.04 | 85.22±0.02 | 92.04±0.05 | 84.98±0.34 | 90.09±0.11 |

While SGCL continues to demonstrate superior performance across most benchmarks in this case, it is noteworthy that SGCL exhibits enhanced performance in the mini-batch scenario compared to the full-batch. We attribute this difference to the presence of a substantial number of connected compo-

nents in the full-batch graph. However, our framework benefits from mini-batch graphs, which can generate *more* connected mini-batches using a random-walk sampling approach. This enables our framework to leverage proximity information within the mini-batch graphs more effectively than in the full-batch graph.

## E.2 Computational analysis

The computational cost of graph contrastive learning models is analyzed through two distinct components: pre-training and downstream task evaluation. In the pre-training phase, the process involves augmentation generation, encoder computation, and computation of the contrastive objective for each batch. In the downstream task phase, the model learns two input/output MLP layers and evaluates the model for node classification. We conduct the computation analysis to evaluate the runtime performance of three variants of the SGCL model and compare these variants with several baseline methods on graphs of different scales, ranging from small to medium and large-scale graphs. The results of these experiments are summarized in Table 7.

Table 7: Runtime performance comparison of the proposed model and baselines across graphs of different scales (seconds).

| Model | Phase | Small (Cora) | Medium (CoauthorCS) | Large (ogbn-arxiv) |
|---|---|---|---|---|
| DGI | pre-training | 0.0391 | 0.0916 | 0.0732 |
| | downstream | 0.0024 | 0.0148 | 0.0837 |
| GRACE | pre-training | 0.0713 | 0.3186 | 0.4233 |
| | downstream | 0.0024 | 0.0148 | 0.0845 |
| MVGRL | pre-training | 0.2266 | 0.7824 | 0.9407 |
| | downstream | 0.0024 | 0.0148 | 0.0833 |
| BGRL | pre-training | 0.0927 | 0.1849 | 0.1755 |
| | downstream | 0.0024 | 0.0149 | 0.0846 |
| GBT | pre-training | 0.0343 | 0.1387 | 0.5388 |
| | downstream | 0.0024 | 0.0148 | 0.0844 |
| SGCL-T | pre-training | 0.1916 | 1.5012 | 2.1334 |
| | downstream | 0.0025 | 0.0149 | 0.0841 |
| SGCL-B | pre-training | 1.3484 | 3.3491 | 3.9549 |
| | downstream | 0.0025 | 0.0151 | 0.0848 |
| SGCL-D | pre-training | 1.3771 | 3.4538 | 4.0496 |
| | downstream | 0.0024 | 0.0151 | 0.0841 |

These results indicate that during pre-training, SGCL-T on the Cora dataset outperforms MVGRL in running time. However, in other experiments, the computational cost of the proposed model is slightly increased compared to the baselines, primarily attributed to the computation associated with the smoothing approach. Specifically, its computational load is approximately twice that of MVGRL. It's noteworthy to highlight that the computational costs in the downstream evaluation phase across all models are nearly identical on each benchmark. This implies that, despite the more computations during the pre-training phase, our model demonstrates efficiency during the downstream evaluation phase.

## F Ablation study

### F.1 Evaluating with other mini-batching generation methods

We conduct node classification experiments employing other mini-batching generation methods, including random node-sampling, random edge-sampling, and Ego-graph. A summary of the results derived from these mini-batching approaches is reported in Table 8.

### F.2 Influence of different terms of contrastive objective

To perform an ablation study on the contrastive objective, we evaluate the significance of each term of Eq. 5 and subsequently combine them with hyperparameter $\lambda$. Table 9 provides the accuracies of different variants of SGCL achieved by different components of the contrastive objective on three benchmarks of varying scales: small (Cora), medium (CoauthorCS), and large (ogbn-arxiv).

Table 8: Accuracy comparison of proposed models with various mini-batching generation approaches (mean ± std).

| Model | Sampling method | Cora | Citeseer | Pubmed | CoauthorCS | Computers | Photo |
|---|---|---|---|---|---|---|---|
| SGCL-T | RW-sampler | **86.54±1.4** | **73.71±2.0** | 86.57±0.49 | 92.99±0.36 | **87.83±1.4** | 93.05±0.94 |
| | Ego-graph | 84.93±4.2 | 71.20±3.4 | 85.70±0.84 | 93.09±0.37 | 86.38±0.9 | **93.15±0.86** |
| | Node-sampler | 84.12±1.8 | 70.42±3.3 | 85.94±0.49 | 92.87±0.58 | 85.92±0.44 | 92.24±0.65 |
| | Edge-sampler | 85.59±2.1 | 70.66±1.4 | **86.76±0.43** | **93.23±0.86** | 86.28±1.3 | 92.31±1.2 |
| SGCL-B | RW-sampler | **87.50±1.7** | **73.65±1.3** | 86.13±0.61 | 93.15±0.39 | **87.89±0.71** | 92.31±1.3 |
| | Ego-graph | 84.56±1.3 | 72.87±1.7 | 85.88±0.33 | 93.10±0.44 | 86.57±1.2 | **93.05±0.85** |
| | Node-sampler | 84.49±1.1 | 70.84±1.9 | 85.90±1.0 | **93.18±0.4** | 85.68±0.57 | 91.95±1.0 |
| | Edge-sampler | 84.26±1.7 | 71.86±2.2 | **86.19±0.7** | 92.85±0.64 | 84.52±0.58 | 91.66±1.1 |
| SGCL-D | RW-sampler | 85.22±0.79 | **72.66±2.4** | 86.27±0.53 | 93.21±0.4 | **85.47±1.2** | **91.86±0.72** |
| | Ego-graph | 85.00±1.4 | 72.57±2.4 | 85.75±0.98 | 92.93±0.3 | 80.04±0.82 | 89.07±0.71 |
| | Node-sampler | 84.26±1.7 | 72.51±1.1 | 85.18±1.0 | 93.06±0.41 | 78.68±0.64 | 89.33±1.6 |
| | Edge-sampler | **86.47±1.2** | 70.42±0.83 | **86.57±0.7** | **93.26±0.08** | 73.31±0.74 | 90.61±0.32 |

Initially, we observe that the exclusion of any term from the loss function results in deteriorated or collapsed solutions, aligning with our expectations. Subsequently, we investigated the influence of the combination of two individual terms using an optimal value of $\lambda$.

To select the value of $\lambda$, we initially set it as $\lambda = 1/2N$. However, in the experiments, we determined its optimal value through grid search. For instance, on the Photo dataset, the optimal value for $\lambda$ was found to be around $2.3e-4$. This value aligns with our first initialization when considering the batch size of $N = 2000$ in the experiments.

Table 9: Accuracies of different SGCL variants influenced by individual components of the contrastive objective Eq. 5.

| Model | Benchmark | (A) | (B) | $\mathcal{L}_{SGCL}^{(i,j)}$ | ($\lambda$) |
|---|---|---|---|---|---|
| SGCL-T | small (Cora) | 84.13±3.1 | 83.54±1.0 | 86.54±1.4 | (4e-4) |
| | medium (CoauthorCS) | 91.36±0.7 | 90.87±0.4 | 92.99±0.4 | (1e-4) |
| | large (ogbn-arxiv) | 68.92±0.0 | 67.05±0.0 | 69.30±0.5 | (1e-4) |
| SGCL-B | small (Cora) | 85.26±2.9 | 85.54±1.3 | 87.50±1.7 | (4e-4) |
| | medium (CoauthorCS) | 93.04±0.1 | 91.45±0.3 | 93.15±0.4 | (1e-4) |
| | large (ogbn-arxiv) | 68.73±0.3 | 68.29±0.4 | 69.24±0.3 | (1e-4) |
| SGCL-D | small (Cora) | 84.91±1.8 | 82.81±2.1 | 85.22±0.8 | (4e-4) |
| | medium (CoauthorCS) | 92.4±0.52 | 91.09±0.2 | 93.2±0.4 | (1e-4) |
| | large (ogbn-arxiv) | 68.40±0.3 | 68.29±0.3 | 69.03±0.4 | (1e-4) |

(A): $\| \tilde{\Pi}_{pos}^{(i,j)} \odot (1 - \mathbf{C}^{(i,j)}) \|_F^2$

(B): $\| (1 - \tilde{\Pi}_{pos}^{(i,j)}) \odot \mathbf{C}^{(i,j)} \|_F^2$

