# OpenReview forum: "Effective Graph Representation Learning via Smoothed Contrastive Learning"
_ICLR.cc/2024/Conference — Submitted to ICLR 2024_

### Official Review · Reviewer_V43w · 2023-10-17

**Soundness:** 2 fair
**Presentation:** 1 poor
**Contribution:** 2 fair
**Rating:** 3
**Confidence:** 4

**Summary:**

This paper focuses on tackling the equal treatment issue of misclassified false negative nodes in conventional GCL approaches. Specifically, the paper presents a Smoothed Graph Contrastive Learning model which leverages the geometric structure of augmented graphs to exploit proximity information associated with positive/negative pairs in contrastive loss. This enables the significance of node pairs to be adjusted. Furthermore, a graph batch-generating strategy that partitions the given graphs into multiple subgraphs is also proposed to facilitate efficient training in separate batches. Experiments show the superiority of the proposed framework.

**Strengths:**

1. This paper is well-motivated. Equal treatment of misclassified false negative nodes and the lack of a mechanism to differentiate misclassified nodes based on proximity can be harmful to graph contrastive learning.
2. Applying smoothing approaches to pair matrices is novel and interesting.

**Weaknesses:**

1. The writing of the paper should be improved. There are many minor mistakes in the paper:
  - "These methods, including , including Deep Graph Infomax" in paragraph 2 of Section 2.
  - "Therfore" in paragraph 2 of Section 3.1.
  - In caption of Figure 2, g(j) is not a positive pair.
  - "distinguishe" in paragraph 1 of Section 4.2.4.
2. A persuasive demonstration of why misaligning negative pairs is harmful should be provided.
3. The smoothing method only employs the original graph information. However, the node relationship can be highly changed after augmentation. For example, two highly related nodes can be dissimilar when one of them is dropped. In such cases, is the proposal still efficient?
4. The proposal seems incremental - the smoothing technique, loss function and subgraph generating can be easily detached from the framework.
5. Can previous contrastive objectives be used in the proposal? There is a lack of ablation studies to exclude the effect of the proposed contrastive objective.
6. A time analysis should be provided to verify the efficiency of the proposal.

**Questions:**

See Weaknesses.

---

> ### Author Response · Authors · 2023-11-21
> **Part 1**
>
> We would like to thank the reviewer for his/her time and effort spent on evaluating our work and their constructive comments. We find the suggestions to be very helpful in improving the quality of our work, making it clearer and more convincing. We are pleased to hear that the reviewer found our idea novel and interesting and is convinced that the paper is well-motivated.
>
> >The writing of the paper should be improved
>
> We appreciate your insights and acknowledge the need for improvement in the writing of the paper. We will carefully review and address the minor mistakes to enhance the overall quality of the manuscript.
>
> >A persuasive demonstration of why misaligning negative pairs is harmful should be provided
>
> We present the following paragraphs in Section 2.1 to demonstrate the detrimental effects of misaligning negative pairs.
>
> In contrastive learning, the misalignment of negative pairs adversely affects the learning process due to its inadvertent impact on the objective function. Consider the following contrastive loss function, designed for each anchor node  $v_t^{(i)}$ with feature embedding $\mathbf{h}_t^i$.
>
> The objective is to minimize the distance between embeddings of positive pair $\( v_t^{(i)},v_t^{(i)} \)$ and simultaneously maximize the distance between embeddings of negative pairs $\( v_t^{(i)},v_q^{(j)}\)_{q=1,q\neq t}^{N-1}$ (Please see Eq. 1 in the paper).
>
> Misalignment in negative pairs $\(v_t^{(i)},v_k^{(j)}\)$ detrimentally impacts the learning process by introducing errors in the loss computation. The misalignment leads to an undesired increase in the loss, hindering the optimization process. Specifically, the GCL model increases the distance between misaligned negative pairs, and inadvertently separates semantically similar samples, leading to a degradation of overall performance.

---

> ### Author Response · Authors · 2023-11-21
> **Part 2**
>
> >The smoothing method only employs the original graph information. However, the node relationship can be highly changed after augmentation. For example, two highly related nodes can be dissimilar when one of them is dropped. In such cases, is the proposal still efficient?
>
> According to the proposed framework, the smoothing method incorporates graph information from each augmented view. Following the generation of two augmented views, we apply the smoothing method to each view independently, aiming to capture every change in the original graph. We sincerely apologize for any confusion that may have arisen from the initial presentation of the framework. In the new version, we have made efforts to provide a clearer explanation.
>
> >The proposal seems incremental....
>
> Respectfully, we adhere to the common framework established for GCL models. Inspired by the DGI framework that proposed an objective based on MI maximization in the graph domain, the majority of GCL models adopt this consistent framework, which encompasses graph augmentation, decoder computation, contrastive objective computation in the pre-training step, and subsequent evaluation in downstream tasks. However, what distinguishes our proposed framework from other GCL models is the intuitive incorporation of proximity information between nodes in positive and negative pairs.
>
> We would like to emphasize that in the conventional GCL framework, there is a lack of information about the proximity of positive/negative pairs, and all N−1 negative pairs are uniformly treated. In other words, the conventional GCL approach treats all misclassified nodes equally, regardless of whether the misclassification occurs near the true positive or at a significant distance from it (since a 1-hop error is just as “expensive” as a k>>1 hop error).
> A common approach for considering proximity involves computing pairwise distances among all nodes in the graph. However, our approach is more efficient, avoiding the need to compute or store large dense matrices. This efficiency is achieved through the integration of three developed smoothing approaches embedded in our loss function.
>
> >Can previous contrastive objectives be used in the proposal?
>
> We can further generalize the smoothed positive/negative pairs to some of the other contrastive objectives with similar settings, which apply positive/negative pairs at the node level. Additionally, the augmented generation strategy should maintain the nodes from the original graph to serve as corresponding positive pairs. For example, we applied the smoothed positive/negative matrices to the contrastive objective used in the GRACE method. However, considering that GRACE utilizes two types of negative pairs (inter-view and intra-view), we constrained it to only inter-view negative pairs. While results on some benchmarks showed improvements over the original GRACE, our approach consistently outperformed it.
>
> > There is a lack of ablation studies to exclude the effect of the proposed contrastive objective.
>
> To perform an ablation study on the contrastive objective, we evaluate the significance of each individual term and subsequently combine them with hyperparameter $\lambda$.
> The following table provides the accuracies of different variants of SGCL achieved by different components of the contrastive objective on three benchmarks of varying scales: small (Cora), medium (CoauthorCS), and large (ogbn-arxiv). Initially, we observe that the exclusion of any term from our loss function results in deteriorated or collapsed solutions, aligning with our expectations. Subsequently, we investigated the influence of the combination of two individual terms using an optimal value of $\lambda$.
>
> |Model | Benchmark  |  First term | Second-term | Loss (Eq. 5) | ($\lambda)$ |
> |----------|----------|----------|----------|----------|----------|
> | | small (Cora) | 84.13±3.1 | 83.54±1.0 | 86.54±1.4 | (4e-4) |
> |SGCL-T | medium (CoauthorCS) | 91.36±0.7 | 90.87±0.4 | 92.99±0.4 | (1e-4) |
> | | large (ogbn-arxiv) | 68.92±0.0 | 67.05±0.0 | 69.30±0.5 | (1e-4) |
> | | small (Cora) | 85.26±2.9 | 85.54±1.3 | 87.50±1.7 | (4e-4) |
> |SGCL-B | medium (CoauthorCS) | 93.04±0.1 | 91.45±0.3 | 93.15±0.4 | (1e-4) |
> | | large (ogbn-arxiv) | 68.73±0.3 | 68.29±0.4 | 69.24±0.3 | (1e-4) |
> | | small (Cora) | 84.91±1.8 | 82.81±2.1 | 85.22±0.8 | (4e-4) |
> |SGCL-D | medium (CoauthorCS) | 92.4±0.52 | 91.09±0.2 | 93.2±0.4 | (1e-4) |
> | | large (ogbn-arxiv) | 68.40±0.3 | 68.29±0.3 | 69.03±0.4 | (1e-4) |
>
> To select the value of $\lambda$, we initially set it as $\lambda =1/2N$​. However, in the experiments, we determined its optimal value through grid search. For instance, on the Photo dataset, the optimal value for $\lambda$ was found to be around $2.3e−4$. This value aligns with our first initialization when considering the batch size of $ N=2000$ in the experiments.

---

> ### Author Response · Authors · 2023-11-21
> **Part 3**
>
> >A time analysis should be provided to verify the efficiency of the proposal.
>
> The computational cost of graph contrastive learning models is analyzed through two distinct components: pre-training and downstream task evaluation. In the pre-training phase, the process involves augmentation generation, encoder computation, and computation of the contrastive objective for each batch. In the downstream task phase, the model learns two input/output MLP layers and evaluates the model for node classification.
>
> We conducted the computation analysis to evaluate the runtime performance of three variants of the SGCL model and compared these variants with several baseline methods on graphs of different scales, ranging from small to medium and large-scale graphs. The results of these experiments are summarized in the following table.
>
> The table results indicate that during pre-training, SGCL-T on the Cora dataset outperforms MVGRL in running time. However, in other experiments, the computational cost of the proposed model is slightly increased compared to the baselines, primarily attributed to the computation associated with the smoothing approach. Specifically, its computational load is approximately twice that of MVGRL.
> It's noteworthy to highlight that the computational costs in the downstream evaluation phase across all models are nearly identical on each benchmark. This implies that, despite the more computations during the pre-training phase, our model demonstrates efficiency during the downstream evaluation phase.
>
> |Model | Phase | Small (Cora) | Medium (CoauthorCS) | Large (ogbn-arxiv)|
> |----------|----------|----------|----------|----------|
> |DGI | pre-training | 0.0391 | 0.0916 | 0.0732 |
> || downstream | 0.0024 | 0.0148 | 0.0837 |
> |GRACE | pre-training | 0.0713 | 0.3186 | 0.4233 |
> | | downstream | 0.0024 | 0.0148 | 0.0845  |
> |MVGRL | pre-training | 0.2266 | 0.7824 | 0.9407 |
> | | downstream | 0.0024 | 0.0148 | 0.0833  |
> |BGRL | pre-training | 0.0927 | 0.1849 | 0.1755 |
> | | downstream | 0.0024 | 0.0149 | 0.0846
> |GBT | pre-training | 0.0343 | 0.1387 | 0.5388 |
> | | downstream | 0.0024 | 0.0148 | 0.0844 |
> |SGCL-T | pre-training | 0.1916 | 1.5012 | 2.1334 |
>  || downstream | 0.0025 | 0.0149 | 0.0841  |
> |SGCL-B | pre-training | 1.3484 | 3.3491 | 3.9549|
> | | downstream | 0.0025 | 0.0151 | 0.0848  |
> |SGCL-D | pre-training | 1.3771 | 3.4538 | 4.0496 |
> | | downstream | 0.0024 | 0.0151 | 0.0841  |

---

### Official Review · Reviewer_jBXi · 2023-10-30

**Soundness:** 3 good
**Presentation:** 4 excellent
**Contribution:** 3 good
**Rating:** 8
**Confidence:** 2

**Summary:**

The authors propose a method called Smoothed Graph Contrastive Learning (SGCL) that tries to use the graph structure to spread out the weights for positive and negative pairs. The matrix $\tilde\Pi_{pos}^{(i,j)}$ is the smoothed out matrix of positive weights (smoothing is done for example using the graph Laplacian matrix). Positive pairs between graph views $\mathcal{G}^{(i)}, \mathcal{G}^{(j)}$ are encouraged to have embeddings with a cosine similarity close to 1. Negative pairs are encouraged to have orthogonal embeddings.

The main contribution of the paper is the idea to smooth positive/negative weights based on graph structure. Experimental results are compellingly in favor of the proposed method.

**Strengths:**

* The main idea to smooth out weights is simple.
* The proposed method performs really well in experiments.

**Weaknesses:**

None that I can come up with.

**Questions:**

What are the final embedding dimensions in the experiments?

Typo:
* page 3, Section 3.2, second paragraph: ${0,1}$ should be $\{0,1\}$
* Section 4.2.3: Is $\tilde D_{ii}$ the degree + 1?

---

> ### Author Response · Authors · 2023-11-21
>
> We express our gratitude to the reviewer for dedicating time and effort to evaluate our work and providing constructive comments.
>
> >The main idea to smooth out weights is simple.
>
> We would like to strongly emphasize that adapting smoothing correspondences to graph contrastive loss is highly non-trivial. Note that in GCL models, in the absence of labeled information, numerous incongruent nodes for each anchor node are potentially treated as false negatives. Specifically, nodes that are close to the anchor and thus have the potential to be semantically similar are inevitably categorized as negative pairs. However, in the conventional GCL methods, which lack information about the proximity of these nodes, all negative pairs are handled uniformly. In other words, the conventional contrastive learning approach treats all misclassified nodes equally, regardless of whether the misclassification occurs near the true positive or at a significant distance from it.
> As outlined in the paper, our approach addresses this limitation by promoting local consistency. A common approach for considering proximity involves computing pairwise distances among all nodes in the graph. However, our approach is more efficient, avoiding the need to compute or store large dense matrices. We effectively integrated three developed smoothing approaches embedded in our loss function. Subsequently, the loss function intuitively incorporates proximity information between nodes in positive and negative pairs.
>
> >What are the final embedding dimensions in the experiments?
>
> In our experiments, we utilize a 512-hidden channel as the dimension of feature embedding learned using the encoder. Consequently, the final embedding dimension in the pre-training step is 512. However, during the evaluation step in downstream tasks, the final embedding dimension aligns with the number of classes in each benchmark. For instance, in the CoauthorCS dataset, it corresponds to 15.
>
> >Typo
>
> We appreciate your insights and recognize the need for corrections, including typos in the writing of the paper. We will carefully review and address the minor mistakes to enhance the overall quality of the paper.
> $D_{ij}$ is the degree matrix computed using the augmented adjacency matrix $\hat{A}=A+I$, which $\hat{A} $ represents the adjacency matrix with a self-loop for each node.

---

### Official Review · Reviewer_EGjE · 2023-10-31

**Soundness:** 3 good
**Presentation:** 2 fair
**Contribution:** 2 fair
**Rating:** 3
**Confidence:** 4

**Summary:**

This paper introduces Smoothed Graph Contrastive Learning to address the issues of false positives and false negatives in graph contrastive learning. The primary idea is to leverage the structural information of the graph to obtain pairwise proximity information and assign weights to each pair. Experimental results demonstrate the effectiveness of the proposed method.

**Strengths:**

- The approach presented in this paper, using graph structural information to smooth the contrastive learning loss, is both intriguing and innovative.
- The proposed method has a solid theoretical foundation.
- The paper provides ample background knowledge to assist readers who may not be familiar with graph smoothing.

**Weaknesses:**

- The structure and organization of this paper appear quite impractical. The author dedicates a substantial portion of Section 3 to background knowledge, occupying a significant amount of space, and only begins to introduce the proposed method towards the end of Page 5. This has resulted in an insufficiently detailed experimental section. It is advisable for the author to trim down the content in Section 3 and allocate more space to the experimental aspects of the paper.
- The author fails to provide the rationale and intuition behind using the loss function as depicted in Eq.4. This loss function does not appear to be particularly innovative. Additionally, I believe that the choice of lambda is crucial, but the author does not explain how lambda is selected.
- The experiments in this paper seem overly simplified, and the dataset splits chosen do not align with commonly used splits in self-supervised learning (public split). The selection of baselines appears outdated, and the reported results in the paper do not align with the results reported for these baseline methods in their original sources.

**Questions:**

I have some questions about the definition of Equation 4.

- In Equation 4, the author claims that $C$ is the cross-correlation matrix of the embedding matrix. However, according to the definition of cross-correlation, $C$ should be an $F\times F$ matrix rather than an $N\times N$ matrix, which contradicts Equation 5. I would recommend the author to double-check this issue.
- In Equation 4, when $i≠j$ and $\hat{\pi}(i, j) = 1$, Eq.4 assigns a high weight to minimize $c_{ij}$. This seems counterintuitive because $\hat{\pi}(i, j) = 1$ should imply that nodes i and j are very likely to be false negatives, so $c_{ij}$ should be maximized rather than minimized.

---

> ### Author Response · Authors · 2023-11-21
> **The rationale and intuition behind using the loss function**
>
> We express our gratitude to the reviewer for dedicating time and effort to evaluate our work and providing constructive comments. The suggestions are invaluable for enhancing the quality of our work, improving clarity, and making it more convincing. We are pleased to learn that the reviewer finds the theoretical foundation of our model to be novel and is convinced that the proposed method is intriguing and innovative.
>
> >The structure and organization of this paper appear quite impractical
>
> Your insights are appreciated. We moved section 2 (related work) to the appendix and revised the paper accordingly to ensure a better balance between sections which allocates more space to the experimental aspects as suggested.
>
> >The author fails to provide the rationale and intuition behind using the loss function
>
> and
>
> >Question1: .... I would recommend the author to double-check this issue.
>
> As per your insightful recommendation, matrix $C \in \{0,1\}^{(N\times N)}$ accurately represents the similarity between node pairs in two augmented graphs, despite our inadvertent reference to it as a cross-correlation matrix.
> We sincerely apologize for any confusion resulting from the unclear presentation and typographical error in Eq. 4. In the first term of this equation, please note that $\textbf{I}$ should be substituted with the scalar "1," and the operator between matrices should be interpreted as the **element-wise** operator.
>
> Kindly refer to the accurate formulation (Eq. 5 in the new paper) provided below:
>
> $L_{SGCL}^{(i,j)} ={ \parallel {\tilde{\Pi}_{p}^{(i,j)}\odot (1-\mathbf{C}^{(i,j)})} \parallel_F^2 +\lambda \parallel (1-{\tilde{\Pi}_p^{(i,j)}})\odot \mathbf{C}^{(i,j)} \parallel_F^2 }$
>
> It appears to us that the rationale behind utilizing the loss function has become more insightful now.
>
> The intuition is straightforward. Generally, matrix $C^{(i,j)}$ comprises similarity values between pairs of nodes from two augmented graphs. Our expectation is to maximize $C^{(i,j)}$ for positive pairs and minimize $C^{(i,j)}$​ for negative pairs, equivalent to simultaneously minimizing $1−C^{(i,j)}​$ for positive pairs and $C^{(i,j)}$​ for negative pairs.
>
> The first term minimizes the discrepancy between '1' and each element of $C^{(i,j)}$, aligning with the values in the smoothed positive pairs matrix $\hat{\Pi}_{p}^{(i,j)}$. This term enforces stability and preservation in the embeddings of positive pairs.
>
> Likewise, the second term minimizes each element of $C^{(i,j)}$ in accordance with the values in the smoothed negative pairs matrix  $\hat{\Pi}_{n}^{(i,j)}=1-\hat{\Pi}_p^{(i,j)}$. This term actively promotes a substantial diversity in the embeddings of negative pairs.
>
>
> >Questions 2:
>
> According to our definition, $\hat{\pi}_{p} (i,j) = 1$ for each $i=j$.
>
> To clarify, if we hypothetically consider that two nodes $i$ and $j$ ($i \neq j$) are very likely to be false negatives, such that $\hat{\pi}_{p}(i,j) = 1$, then the first term assigns a weight of '1' to minimize $1 - C^{(i,j)}$ which is equivalent to maximizing $ C^{(i,j)}$, as you rightly mentioned. However, the second term is redundant since it assigns a weight of '0' to minimize $ C^{(i,j)}$.

---

> ### Author Response · Authors · 2023-11-21
>
> >I believe that the choice of lambda is crucial, but the author does not explain how lambda is selected.
>
> To select the value of $\lambda$, we initially set it as $\lambda =1/2N$​. However, in the experiments, we determined its optimal value through grid search. For instance, on the Photo dataset, the optimal value for $\lambda$ was found to be around $2.3e−4$. This value aligns with our first initialization when considering the batch size of $N=2000$ in the experiments.
>
> Additionally, we performed an ablation study on the contrastive objective. We evaluated the significance of each individual term and subsequently combined them with hyperparameter $\lambda$.
> The following table provides the accuracies of different variants of SGCL achieved by different components of the contrastive objective on three benchmarks of varying scales: small (Cora), medium (CoauthorCS), and large (ogbn-arxiv). Initially, we observe that the exclusion of any term from our loss function results in deteriorated or collapsed solutions, aligning with our expectations. Subsequently, we investigated the influence of the combination of two individual terms using an optimal value of $\lambda$.
>
> |Model | Benchmark  |  First term | Second-term | Loss (Eq. 5) | ($\lambda)$ |
> |----------|----------|----------|----------|----------|----------|
> | | small (Cora) | 84.13±3.1 | 83.54±1.0 | 86.54±1.4 | (4e-4) |
> |SGCL-T | medium (CoauthorCS) | 91.36±0.7 | 90.87±0.4 | 92.99±0.4 | (1e-4) |
> | | large (ogbn-arxiv) | 68.92±0.0 | 67.05±0.0 | 69.30±0.5 | (1e-4) |
> | | small (Cora) | 85.26±2.9 | 85.54±1.3 | 87.50±1.7 | (4e-4) |
> |SGCL-B | medium (CoauthorCS) | 93.04±0.1 | 91.45±0.3 | 93.15±0.4 | (1e-4) |
> | | large (ogbn-arxiv) | 68.73±0.3 | 68.29±0.4 | 69.24±0.3 | (1e-4) |
> | | small (Cora) | 84.91±1.8 | 82.81±2.1 | 85.22±0.8 | (4e-4) |
> |SGCL-D | medium (CoauthorCS) | 92.4±0.52 | 91.09±0.2 | 93.2±0.4 | (1e-4) |
> | | large (ogbn-arxiv) | 68.40±0.3 | 68.29±0.3 | 69.03±0.4 | (1e-4) |

---

> ### Author Response · Authors · 2023-11-21
> **Experiments**
>
> >The experiments in this paper seem overly simplified...
>
> In our experiments, we employed the PyGCL library to explore a mini-batch scenario, aligning with the setting of our proposed model. Considering the diverse settings across various models developed for contrastive learning and the absence of comprehensive experiments on several Ogbn benchmarks in the published paper, to ensure fair comparisons, we reproduced the results of the state-of-the-art models available within this library, as detailed in Table 1 of the paper.  We will publish the code of our model as well as the SOTA models for easy reproducibility of the results once the paper is accepted.
>
> For further investigation, we conducted additional experiments involving a range of state-of-the-art methods, notably CGRA [1] and GRLC [2]. To establish a fair and robust comparison with the existing SOTA methods, these experiments are conducted in a full-batch scenario, adhering to the **commonly employed data split in self-supervised learning** as provided in the OGB benchmarks. The results are summarized in the following table.
>
> [1 H. Duan et al., "Self-supervised contrastive graph representation with node and graph augmentation," Neural Networks, (67), 2023
>
> [2] L. Peng et al., "GRLC: Graph Representation Learning With Constraints," in IEEE Transactions on Neural Networks and Learning Systems, 2023
>
> |Model | Cora | Citeseer | Pubmed | CoauthorCS | Computers | Photo |
> |---------- | ---------- | ---------- | ---------- | ---------- | ---------- | ---------- |
> |DGI   | 76.28±0.04 | 69.33±0.14 | 83.79±0.08 | 91.63±0.08 | 71.96±0.06 | 75.27±0.02 |
> |GRACE  | 81.80±0.19 | 71.35±0.07 | 85.86±0.05 | 91.57±0.14 | 84.77±0.06 | 89.50±0.06|
> |MVGRL  | 84.98±0.11 | 71.29±0.04 | 85.22±0.04 | 91.65±0.02 | 88.55±0.02 | 91.90±0.08 |
> |BGRL  | 80.21±1.14 | 66.33±2.10 | 81.78±1.06 | 90.19±0.82 | 84.24±1.32 | 89.56±1.01 |
> |GBT | 79.32±0.31 | 65.78±1.33 |**86.35±0.48** | 91.87±0.07 | **90.43±0.18** | 92.23±0.18 |
> |CGRA  |82.71±0.01  |69.23±1.19  |82.15±0.46  |91.26±0.27  |89.76±0.36  |91.54±1.06|
> |GRLC  |83.50±0.24 | 70.02±0.16 | 81.20±0.20 | 90.36±0.27 | 88.54±0.23 | 91.80±0.77|
> |**SGCL-T** | 84.45±0.04| 71.26±0.06 |84.11±0.08 |92.14±0.09 | 86.81±0.01 |**92.71±0.05** |
> |**SGCL-B** | **85.08±0.12**| **72.77±0.33** |83.67±0.06 |    **92.16±0.15** | 88.24±0.05| 92.43±0.03 |
> |**SGCL-D** | 84.47±0.25| 70.32±0.04 |85.22±0.02 |    92.04±0.05 |84.98±0.34 |90.09±0.11 |
>
> Additionally, we have extended the evaluation of the model by conducting new experiments for the graph classification task, using five commonly used graph classification benchmarks: MUTAG, PTC, IMDB-BINARY, PROTEINS, and ENZYMES. In this experiment, we followed the InfoGraph setting for graph classification and compared the accuracy with the self-supervised state-of-the-art methods, including InfoGraph, GraphCL, MVGRL, BGRL, AD-GCL, LaGraph, and CGRA (Please see section 4.3).
>
> The results reported in the following table (Table 3 in the paper) indicate that, in comparison to the best-performing state-of-the-art methods, the proposed approach demonstrates enhanced accuracy for IMDB-BINARY, PROTEINS, and ENZYMES, while maintaining comparable accuracy on other benchmarks.
>
> It's worth mentioning that the accuracies of all models are reported from their respective published papers, except for the BGRL results, which we reproduced under the same experimental setting.
>
> Model | IMDB-Binary | PTC | MUTAG | PROTEINS | ENZYMES
> |----------|----------|----------|----------|----------|----------|
> |InfoGraph | 73.0±0.9 | 61.7±1.4 | 89.0±1.1 | 74.4±0.3 | 50.2±1.4 |
> |GraphCL | 71.1±0.4 | 63.6±1.8 | 86.8±1.3 | 74.4±0.5 | 55.1±1.6 |
> |MVGRL  | 74.2±0.7 | 62.5±1.7 | 89.7±1.1 | 71.5±0.3 | 48.3±1.2 |
> |AD-GCL  | 71.5±1.0 | 61.2±1.4 | 86.8±1.3 | 75.0±0.5 | 42.6±1.1 |
> |BGRL  | 72.8±0.5 | 57.4±0.9 | 86.0±1.8 | 77.4±2.4 | 50.7±9.0 |
> |LaGraph  | 73.7±0.9 | 60.8±1.1 | 90.2±1.1 | 75.2±0.4 | 40.9±1.7|
> |CGRA  | 75.6±0.5 | **65.7±1.8** | **91.1±2.5** | 76.2±0.6 | 61.1±0.9 |
> |**SGCL-T** | 75.2±2.8 | 64.0±1.6 | 89.0±2.3 | 79.4±1.9 | **65.3±3.6**|
> |**SGCL-B** | 73.2±3.7 | 62.5±1.8 | 87.0±2.8 | **81.6±2.3** | 63.7±1.6 |
> |**SGCL-D** | **75.8±1.9** | 62.6±1.4 | 86.0±2.6 | 81.5±2.3 | 64.3±2.2 |

---

### Official Review · Reviewer_4ysC · 2023-11-01

**Soundness:** 2 fair
**Presentation:** 2 fair
**Contribution:** 2 fair
**Rating:** 5
**Confidence:** 3

**Summary:**

The paper tackles the challenge in Graph contrastive learning (GCL), particularly the problem of uniformly incorporating negative samples in the contrastive loss, which may not account for the proximity of the true positive nodes. The authors introduced a new method Smoothed Graph Contrastive Learning model (SGCL), aiming to consider the geometric structure of augmented graphs and exploit proximity information for better representation learning.

**Strengths:**

1. The presentation is clear.
2. The studied problem is interesting.

**Weaknesses:**

1. The newest baseline is published in 2022. Therefore the paper misses a lot SOTA methods.
2. The paper doesn't involve computational cost analysis. Moreover, the cost should be compared with baselines.
3. Smoothing for graph contrastive learning seems to be a little trivial to me.
4. Can the proposed methods be adopted for graph-level tasks [1,2]?

[1] Supervised Contrastive Learning with Structure Inference for Graph Classification, TNNLS

[2] Self-supervised Graph-level Representation Learning with Adversarial Contrastive Learning, TKDD

**Questions:**

See above

---

> ### Author Response · Authors · 2023-11-21
> **New experiment with newest baselines**
>
> We would like to thank the reviewer for his/her time and effort spent on evaluating our work and their constructive comments. We find the suggestions to be very helpful for improving the quality of our work, making it more convincing. We are pleased to learn that the reviewer is interested in the problem and has found the presentation of our method to be clear.
>
> >The newest baseline is published in 2022. Therefore the paper misses a lot SOTA methods.
>
> In our experiments, we employed the PyGCL library to explore a mini-batch scenario, aligning with the setting of our proposed model (https://github.com/PyGCL/PyGCL). Considering the diverse settings across various models developed for contrastive learning and the absence of comprehensive experiments on several Open Graph Benchmarks in the published paper, to ensure fair comparisons, we reproduced the results of the state-of-the-art models available within this library, as detailed in Table 1 of the paper. We will publish the code of our model as well as the SOTA models for easy reproducibility of the results once the paper is accepted.
> For further investigation, we conducted additional experiments involving a range of state-of-the-art methods, notably CGRA [1] and GRLC [2]. To establish a fair and robust comparison with the existing SOTA methods, these experiments are conducted in a full-batch scenario, adhering to the **commonly employed data split in self-supervised learning as provided in the OGB benchmarks**. The results are summarized in the following table.
>
> [1] H. Duan et al., "Self-supervised contrastive graph representation with node and graph augmentation," Neural Networks, (67), 2023
>
> [2] L. Peng et al., "GRLC: Graph Representation Learning With Constraints," in IEEE Transactions on Neural Networks and Learning Systems, 2023
>
> |Model | Cora | Citeseer | Pubmed | CoauthorCS | Computers | Photo |
> |---------- | ---------- | ---------- | ---------- | ---------- | ---------- | ---------- |
> |DGI   | 76.28±0.04 | 69.33±0.14 | 83.79±0.08 | 91.63±0.08 | 71.96±0.06 | 75.27±0.02 |
> |GRACE  | 81.80±0.19 | 71.35±0.07 | 85.86±0.05 | 91.57±0.14 | 84.77±0.06 | 89.50±0.06|
> |MVGRL  | 84.98±0.11 | 71.29±0.04 | 85.22±0.04 | 91.65±0.02 | 88.55±0.02 | 91.90±0.08 |
> |BGRL  | 80.21±1.14 | 66.33±2.10 | 81.78±1.06 | 90.19±0.82 | 84.24±1.32 | 89.56±1.01 |
> |GBT | 79.32±0.31 | 65.78±1.33 |**86.35±0.48** | 91.87±0.07 | **90.43±0.18** | 92.23±0.18 |
> |CGRA  |82.71±0.01  |69.23±1.19  |82.15±0.46  |91.26±0.27  |89.76±0.36  |91.54±1.06|
> |GRLC  |83.50±0.24 | 70.02±0.16 | 81.20±0.20 | 90.36±0.27 | 88.54±0.23 | 91.80±0.77|
> |**SGCL-T** | 84.45±0.04| 71.26±0.06 |84.11±0.08 |92.14±0.09 | 86.81±0.01 |**92.71±0.05** |
> |**SGCL-B** | **85.08±0.12**| **72.77±0.33** |83.67±0.06 |    **92.16±0.15** | 88.24±0.05| 92.43±0.03 |
> |**SGCL-D** | 84.47±0.25| 70.32±0.04 |85.22±0.02 |    92.04±0.05 |84.98±0.34 |90.09±0.11 |

---

> ### Author Response · Authors · 2023-11-21
> **Computational cost analysis**
>
> The computational cost of graph contrastive learning models is analyzed through two distinct components: pre-training and downstream task evaluation. In the pre-training phase, the process involves augmentation generation, encoder computation, and computation of the contrastive objective for each batch. In the downstream task phase, the model learns two input/output MLP layers and evaluates the model for node classification.
>
> We conducted the computation analysis to evaluate the runtime performance of three variants of the SGCL model and compared these variants with several baseline methods on graphs of different scales, ranging from small to medium and large-scale graphs. The results of these experiments are summarized in the following table.
>
> The table results indicate that during pre-training, SGCL-T on the Cora dataset outperforms MVGRL in running time. However, in other experiments, the computational cost of the proposed model is slightly increased compared to the baselines, primarily attributed to the computation associated with the smoothing approach. Specifically, its computational load is approximately twice that of MVGRL.
> It's noteworthy to highlight that the computational costs in the downstream evaluation phase across all models are nearly identical on each benchmark. This implies that, despite the more computations during the pre-training phase, our model demonstrates efficiency during the downstream evaluation phase.
>
> |Model | Phase | Small (Cora) | Medium (CoauthorCS) | Large (ogbn-arxiv)|
> |----------|----------|----------|----------|----------|
> |DGI | pre-training | 0.0391 | 0.0916 | 0.0732 |
> || downstream | 0.0024 | 0.0148 | 0.0837 |
> |GRACE | pre-training | 0.0713 | 0.3186 | 0.4233 |
> | | downstream | 0.0024 | 0.0148 | 0.0845  |
> |MVGRL | pre-training | 0.2266 | 0.7824 | 0.9407 |
> | | downstream | 0.0024 | 0.0148 | 0.0833  |
> |BGRL | pre-training | 0.0927 | 0.1849 | 0.1755 |
> | | downstream | 0.0024 | 0.0149 | 0.0846
> |GBT | pre-training | 0.0343 | 0.1387 | 0.5388 |
> | | downstream | 0.0024 | 0.0148 | 0.0844 |
> |SGCL-T | pre-training | 0.1916 | 1.5012 | 2.1334 |
>  || downstream | 0.0025 | 0.0149 | 0.0841  |
> |SGCL-B | pre-training | 1.3484 | 3.3491 | 3.9549|
> | | downstream | 0.0025 | 0.0151 | 0.0848  |
> |SGCL-D | pre-training | 1.3771 | 3.4538 | 4.0496 |
> | | downstream | 0.0024 | 0.0151 | 0.0841  |

---

> ### Author Response · Authors · 2023-11-21
> **Contrastive learning model**
>
> >Smoothing for graph contrastive learning seems to be a little trivial to me.
>
> We would like to strongly emphasize that adapting smoothing correspondences to graph contrastive loss is highly non-trivial. Note that in GCL models, in the absence of labeled information, numerous incongruent nodes for each anchor node are potentially treated as false negatives. Specifically, nodes that are close to the anchor and thus have the potential to be semantically similar are inevitably categorized as negative pairs. However, in the conventional GCL methods, which lack information about the proximity of these nodes, all negative pairs are handled uniformly. In other words, the conventional contrastive learning approach treats all misclassified nodes equally, regardless of whether the misclassification occurs near the true positive or at a significant distance from it.
>
> As outlined in the paper, our approach addresses this limitation by promoting local consistency. A common approach for considering proximity involves computing pairwise distances among all nodes in the graph. However, our approach is more efficient, avoiding the need to compute or store large dense matrices. We effectively integrated three developed smoothing approaches embedded in our loss function. Subsequently, the loss function intuitively incorporates proximity information between nodes in positive and negative pairs.

---

> ### Author Response · Authors · 2023-11-21
> **graph-level tasks**
>
> >Can the proposed methods be adopted for graph-level tasks [1,2]?
>
> We acknowledge that our proposed model is able to be adopted for graph-level tasks. We have evaluated the proposed model for the graph classification task using five commonly used graph classification benchmarks: MUTAG, PTC, IMDB-BINARY, PROTEINS, and ENZYMES. In this experiment, we followed the InfoGraph setting for graph classification and compared the accuracy with the self-supervised state-of-the-art methods, including InfoGraph, GraphCL, MVGRL, BGRL, AD-GCL, LaGraph, and CGRA.
>
> The results reported in the following table (Table 3 in the paper)  indicate that, in comparison to the best-performing state-of-the-art methods, the proposed approach demonstrates enhanced accuracy for IMDB-BINARY, PROTEINS, and ENZYMES, while maintaining comparable accuracy on other benchmarks.
> It's worth mentioning that the accuracies of all models are reported from their respective published papers, except for the BGRL results, which we reproduced under the same experimental setting.
>
> Model | IMDB-Binary | PTC | MUTAG | PROTEINS | ENZYMES
> |----------|----------|----------|----------|----------|----------|
> |InfoGraph | 73.0±0.9 | 61.7±1.4 | 89.0±1.1 | 74.4±0.3 | 50.2±1.4 |
> |GraphCL | 71.1±0.4 | 63.6±1.8 | 86.8±1.3 | 74.4±0.5 | 55.1±1.6 |
> |MVGRL  | 74.2±0.7 | 62.5±1.7 | 89.7±1.1 | 71.5±0.3 | 48.3±1.2 |
> |AD-GCL  | 71.5±1.0 | 61.2±1.4 | 86.8±1.3 | 75.0±0.5 | 42.6±1.1 |
> |BGRL  | 72.8±0.5 | 57.4±0.9 | 86.0±1.8 | 77.4±2.4 | 50.7±9.0 |
> |LaGraph  | 73.7±0.9 | 60.8±1.1 | 90.2±1.1 | 75.2±0.4 | 40.9±1.7|
> |CGRA  | 75.6±0.5 | **65.7±1.8** | **91.1±2.5** | 76.2±0.6 | 61.1±0.9 |
> |**SGCL-T** | 75.2±2.8 | 64.0±1.6 | 89.0±2.3 | 79.4±1.9 | **65.3±3.6**|
> |**SGCL-B** | 73.2±3.7 | 62.5±1.8 | 87.0±2.8 | **81.6±2.3** | 63.7±1.6 |
> |**SGCL-D** | **75.8±1.9** | 62.6±1.4 | 86.0±2.6 | 81.5±2.3 | 64.3±2.2 |
>
> We would like to mention that among the two references you recommended, we compared the accuracies using the second reference, CGRA, while the first reference was supervised, making the comparison inherently unfair.

---

### Author Response · Authors · 2023-11-21

We would like to thank the reviewers for their time and effort spent evaluating our work and their constructive comments. We find the suggestions to be very helpful in improving the quality of our work, making it clearer and more convincing. We are pleased to hear that most of the reviewers found our paper well-written and are convinced that the proposed method is well-analyzed and interesting.

Before addressing individual comments, we stress the following contributions of our work:

1. We explained, for the first time, that conventional GCL approaches lack a mechanism to differentiate and adequately penalize misclassified nodes. In GCL approaches there are a significant number of negative pairs that have the potential for false negatives. Specifically, conventional GCL  approaches allocate negative pairs between views uniformly without taking into account the fact that the misclassified nodes closer to the positive node should incur a lower penalty compared to those located farther away (since a 1-hop error is just as “expensive” as a k>>1 hop error). Our approach helps to overcome this limitation by promoting local consistency. A common approach for considering proximity involves computing pairwise distances among all nodes in the graph. However, our approach is more efficient, avoiding the need to compute or store large dense matrices. This efficiency is achieved through the integration of three developed smoothing approaches embedded in our loss function.

2. We introduced and compared three alternative formulations for incorporating smoothing into our contrastive learning objective function. These formulations intuitively utilize three distinct strategies to incorporate proximity information when assigning positive and negative pairs.

3. We introduced a novel graph contrastive objective function that incorporates computed smoothed matrices. It allows us to align different positive and negative terms in a flexible manner, while directly incorporating our smoothing approach.

4. We performed experiments for both node and graph classification on a diverse set of standard graph benchmarks of different scales (small, medium, and large) using two scenarios: full-batch and mini-batch sampling. For the ablation study, we performed four batch-generating approaches and also studied the importance of different terms in the proposed SGCL loss function.

5. To evaluate the performances, we compared the accuracy with a variety of state-of-the-art methods.  To ensure a fair and robust comparison with the existing state-of-the-art methods, we reproduced the results of all SOTA models.

Moreover, we would like to strongly emphasize that adapting smoothing correspondences to graph contrastive loss is highly non-trivial. Note that in GCL models, in the absence of labeled information, numerous incongruent nodes for each anchor node are potentially treated as false negatives. Specifically, nodes that are close to the anchor and thus have the potential to be semantically similar are inevitably categorized as negative pairs. However, in the conventional GCL methods, which lack information about the proximity of these nodes, all negative pairs are handled uniformly. In other words, the conventional contrastive learning approach treats all misclassified nodes equally, regardless of whether the misclassification occurs near the true positive or at a significant distance from it.

As outlined in the paper, our approach addresses this limitation by promoting local consistency. We effectively integrate the geometric structure of graph views into a smoothed contrastive loss function. Subsequently, the loss function intuitively incorporates proximity information between nodes in positive and negative pairs.

---

### Meta-Review · Area_Chair_fgoE · 2023-12-08

**Metareview:**

There were several major concerns raised by the reviewers in their original reviews, including 1) the time efficiency comparison between the proposed method and baselines is missing; 2, the baselines are not SOTA; 3, the method is incremental in terms of technical novelty; 4, the intuition of the loss function is not clear; etc. During rebuttal, though some concerns are partially addressed, major concerns remain: the proposed method is a combination of existing techniques; the time analysis is not convincing, as it only shows the computation time for each small batch, not an epoch or the whole training time, the reported time for downstream inference tasks is confusing, is it for the whole test set or a single graph? the explanations of the intuition of the loss function are still not clear or convincing.

Based on the current shape, this work is not ready for publication.

**Justification For Why Not Higher Score:**

There are quite a number of major concerns.

**Justification For Why Not Lower Score:**

N/A

---

### Decision · Program_Chairs · 2024-01-16

Reject